# DAT: Improving Adversarial Robustness via Generative Amplitude Mix-up in Frequency Domain

**Fengpeng Li[1]**    **Kemou Li[1]**    **Haiwei Wu[2]**    **Jinyu Tian[3]**    **Jiantao Zhou[1]**[*]

[1]State Key Laboratory of Internet of Things for Smart City, University of Macau
[2]Department of Computer Science, City University of Hong Kong
[3]Faculty of Innovation Engineering, Macau University of Science and Technology

## Abstract

To protect deep neural networks (DNNs) from adversarial attacks, adversarial training (AT) is developed by incorporating adversarial examples (AEs) into model training. Recent studies show that adversarial attacks disproportionately impact the patterns within the phase of the sample's frequency spectrum—typically containing crucial semantic information—more than those in the amplitude, resulting in the model's erroneous categorization of AEs. We find that, by mixing the amplitude of training samples' frequency spectrum with those of distractor images for AT, the model can be guided to focus on phase patterns unaffected by adversarial perturbations. As a result, the model's robustness can be improved. Unfortunately, it is still challenging to select appropriate distractor images, which should mix the amplitude without affecting the phase patterns. To this end, in this paper, we propose an optimized *Adversarial Amplitude Generator (AAG)* to achieve a better tradeoff between improving the model's robustness and retaining phase patterns. Based on this generator, together with an efficient AE production procedure, we design a new *Dual Adversarial Training (DAT)* strategy. Experiments on various datasets show that our proposed DAT leads to significantly improved robustness against diverse adversarial attacks. The source code is available at `https://github.com/Feng-peng-Li/DAT`.

## 1  Introduction

DNNs have been successfully applied to various tasks [21, 32, 25]. However, recent studies reveal that DNNs are vulnerable to adversarial examples (AEs), created by applying subtle yet deceptive adversarial perturbations to benign samples [40, 51]. Such vulnerabilities have sparked considerable interests, leading to the development of numerous adversarial attacks designed to deceive DNNs [40, 19, 39, 13, 20, 5, 6]. Furthermore, serious concerns about the trustworthiness of artificial intelligence have been raised, due to these fundamental vulnerabilities [47, 37]. To mitigate these threats, adversarial training (AT) has been developed to enhance model robustness by incorporating AEs into training through a min-max strategy [40, 61, 38]. Based on the typical method, PGD-AT [40], a variety of AT strategies have been devised [38, 29, 37] (see Appendix B for related works).

For image signals transformed into the frequency domain using, *e.g.* the discrete Fourier transform (DFT), several AT works [59, 52, 53, 43] explore the adversarial attacks' behavior on sample's frequency spectrum. Frequency spectra consist of amplitude and phase; the amplitude typically captures stylistic information, whereas the phase encompasses richer semantics [7]. Recent studies [58, 63] find, as shown in Figure 1, that adversarial attacks often severely eliminate some semantics in the phase, making it difficult for models to extract features for correctly predicting AEs, while the

---

[*]Corresponding author: Jiantao Zhou (jtzhou@um.edu.mo).

38th Conference on Neural Information Processing Systems (NeurIPS 2024).

impacts on amplitude are relatively mild. Moreover, by forcing the model to focus on phase patterns, [7, 63] confirm the model can learn more features unaffected by image corruptions, *e.g.*, Gaussian noise and defocus blur, improving the model's performance on corrupted samples.

To explore the impact of adversarial perturbations on phase and amplitude patterns respectively, we conduct some studies (see Sec. 2 for details). We find that the standard model trained without AT has worse performance on samples with adversarial phase patterns than those with adversarial amplitude ones. Unlike the standard model, AT enhances the model's robustness against both phase- and amplitude-level adversarial perturbations, with a more noticeable improvement against adversarial perturbations on phase patterns. This indicates the potential for improving the model's robustness by focusing on phase patterns unaffected by adversarial perturbations in AT. Then, by mixing the training samples' amplitude with randomly selected distractor, we observe that the robust model performance, particularly at adversarial phase patterns, is further enhanced, without impacting that at adversarial amplitude ones. This demonstrates that training samples with mixed amplitude improve the model's performance on AEs, and maintain the model's robustness on amplitude-level.

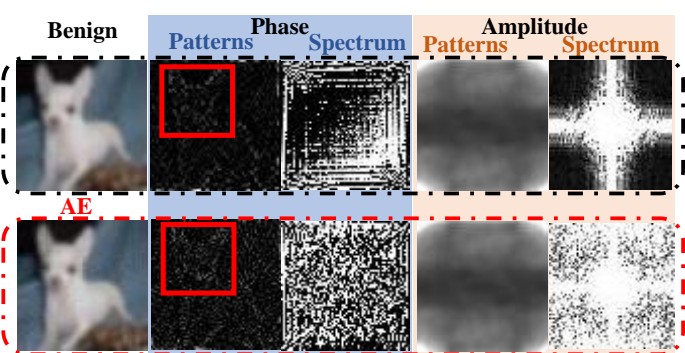

Figure 1: The adversarial perturbation severely damages phase patterns (especially in red rectangular) and the frequency spectrum, while amplitude patterns are rarely impacted. The AE is generated by PGD-20 $\ell_\infty$-bounded with radius $8/255$.

Inspired by these observations, we in this work propose a new *Dual Adversarial Training (DAT)* strategy by focusing on the phase patterns, with two adversarial procedures: adversarial amplitude generation and efficient AE production. Specifically, to guide the model to learn more phase patterns, we first attempt to mix the amplitude of training samples' frequency spectra with randomly selected distractor images. However, when the disparity between the distractor and the original image is too large, the recombined ones tend to disrupt original phase patterns, hindering the model from predicting AE correctly. Conversely, it is difficult for models to focus on phase patterns when the distractor closely resembles the original one [7]. To tackle this challenge, we propose an optimized *Adversarial Amplitude Generator (AAG)* to synthesize an adversarial amplitude, maximizing the model loss and limiting the model fitting amplitude patterns, thereby the model focusing on phase patterns for convergence. During the training process, the AAG and robust model are optimized jointly with the original and recombined images, together with their AEs. The robust model undergoes training by empirical risk minimization, and maximizes the model loss to update the AAG adversarially. Experiments across various benchmarks against a range of adversarial attacks confirm the superior effectiveness of our proposed DAT, surpassing state-of-the-art methods in robustness with big margins.

**Contribution.** The contributions of this work can be summarized as:

- We verify that adversarial perturbations significantly influence phase patterns, resulting in the model's difficulty for predicting AEs correctly. Moreover, by mixing the amplitude of a training image with that of a distractor, we find that the model robustness against AEs can be enhanced.

- We propose the novel DAT strategy with an optimized AAG to synthesize an adversarial amplitude. With the AAG, we enforce the model to better focus on phase patterns, enhancing the model's robustness. Also, an efficient AE generation module is incorporated to improve the AT's efficiency.

- Experiments on multiple datasets confirm that DAT significantly enhances the model's robustness against a variety of adversarial attacks. Specifically, DAT increases the model's robustness by $\sim$2.1% on CIFAR-10, $\sim$2.2% on CIFAR-100, and $\sim$2.3% on Tiny ImageNet, on average.

**Notation.** Let $\mathcal{D} = \{(\mathbf{x}_i, y_i)\}_{i=1}^N$ be a benign dataset comprising $N$ samples from $c$ classes, where each sample $\mathbf{x}_i \in \mathcal{X} \subseteq \mathbb{R}^{C \times H \times W}$ is an image with $C$ channels, height $H$, and width $W$, and its label $y_i \in [c] = \{1, \ldots, c\}$. $f_{\boldsymbol{\theta}} : \mathcal{X} \to \mathbb{R}^c$ denotes a DNN function parameterized by $\boldsymbol{\theta} \in \mathbb{R}^d$,

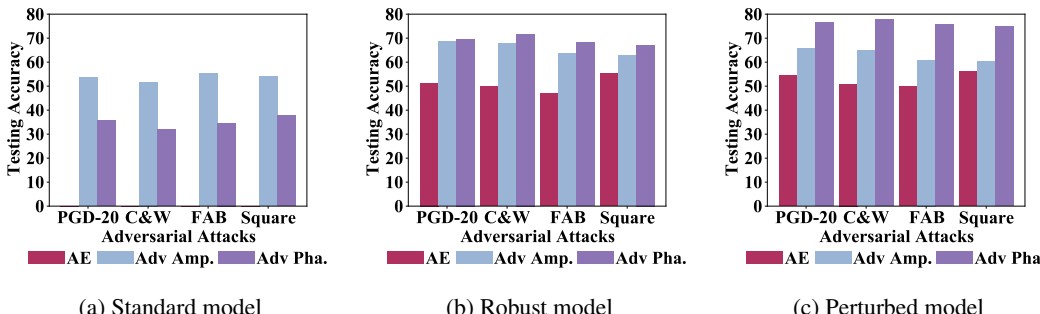

|  (a) Standard model  |  (b) Robust model  |  (c) Perturbed model  |

Figure 2: Test accuracy (%) on CIFAR-10 of (a) standard, (b) robust, and (c) amplitude-perturbed ResNet-18. "Adv Amp./Pha." refers to $\mathbf{x}'_{amp}/\mathbf{x}'_{pha}$. AEs are generated by PGD-20, C&W$_\infty$, FAB, and Square, with $\ell_\infty$-bounded perturbation budget $\epsilon = 8/255$ and inner step $\alpha = 2/255$. The robust and perturbed models are trained by PGD-AT-10 following [40].

and $F_{\boldsymbol{\theta}}(\mathbf{x}) = \arg\max_{y \in [c]} f_{\boldsymbol{\theta}}(\mathbf{x})_y$ is the predicted label of $\mathbf{x}$. Moreover, $f_{\boldsymbol{\theta}} = g \circ h$, where $h$ is the feature extractor and $g$ is the classifier. Let $\mathcal{H} \subseteq \mathbb{R}^m$ be the $m$-dimensional feature space, $h_i : \mathcal{X} \to \mathbb{R}$ be a feature mapping function, and $\mathbf{h}(\mathbf{x}) = [h_1(\mathbf{x}), \ldots, h_m(\mathbf{x})]^\top \in \mathcal{H}$ denote the feature map of $\mathbf{x}$. Specifically, features induced from the amplitude and phase patterns of $\mathbf{x}$ are $h_a(\mathbf{x})$ and $h_p(\mathbf{x})$, respectively. Define $\mathcal{S}_\epsilon[\mathbf{x}] = \{\mathbf{x}' : \|\mathbf{x}' - \mathbf{x}\|_\infty \leqslant \epsilon\}$ as an $\ell_\infty$-ball centered on $\mathbf{x}$ with radius $\epsilon$. Let $\mathcal{F}(\cdot)$ and $\mathcal{F}^{-1}(\cdot, \cdot)$ denote the DFT and inverse DFT (IDFT) functions. Typically, DFT is independently applied to each channel of an image $\mathbf{x}$ within the pixel space as:

$$\mathcal{F}(\mathbf{x})(u, v) = \sum_{h=1}^{H} \sum_{w=1}^{W} \mathbf{x}(h, w) \, \mathrm{e}^{-\mathrm{i}2\pi(u\frac{h}{H} + v\frac{w}{W})},$$

where $(h, w)$ denotes the pixel coordinates of $\mathbf{x}$, and $(u, v) \in [H] \times [W]$ signifies coordinates in the frequency domain. The real and imaginary parts of $\mathcal{F}(\mathbf{x})$ are denoted by $\mathrm{Re}(\mathcal{F}(\mathbf{x}))$ and $\mathrm{Im}(\mathcal{F}(\mathbf{x}))$, respectively. Then, the amplitude spectrum $\mathcal{A}(\mathbf{x})$ and phase spectrum $\mathcal{P}(\mathbf{x})$ are

$$\mathcal{A}(\mathbf{x}) = \left(\mathrm{Re}^2(\mathcal{F}(\mathbf{x})) + \mathrm{Im}^2(\mathcal{F}(\mathbf{x}))\right)^{\frac{1}{2}}, \quad \mathcal{P}(\mathbf{x}) = \arctan\left(\frac{\mathrm{Im}(\mathcal{F}(\mathbf{x}))}{\mathrm{Re}(\mathcal{F}(\mathbf{x}))}\right). \tag{1}$$

## 2 Motivation: On Improving Adversarial Robustness in Frequency Domain

To investigate the approach to enhance the model robustness and show the motivation of DAT, we perform exploration experiments in this section. As shown in Figure 1, adversarial perturbations severely compromise the semantics within phase patterns, resulting in the difficulty of the model predicting AEs correctly. Consequently, we examine the distinct effects of adversarial perturbations on amplitude and phase patterns. To achieve this target, we employ the standard and robust models, trained without and with AT as [40] on $\mathcal{D}_t$ (the training subset of CIFAR-10). Moreover, on training samples with perturbed amplitude, the model tends to focus on phase patterns, in order to achieve the convergence [7, 58]. Following this line, we discuss the impact of perturbing amplitude by mixing the amplitude of training samples with those of distractors randomly selected from $\mathcal{D}_t$. For $(\mathbf{x}, y) \in \mathcal{D}_t$, the recombined sample $\hat{\mathbf{x}}$ is generated by amplitude-level mixing operations and IDFT, namely,

$$\hat{\mathbf{x}} = \mathcal{F}^{-1}(\lambda \cdot \mathcal{A}(\mathbf{x}_0) + (1 - \lambda) \cdot \mathcal{A}(\mathbf{x}), \mathcal{P}(\mathbf{x})), \quad \lambda \sim \mathrm{Uniform}(0, 1), \tag{2}$$

where $\mathbf{x}_0$ is the distractor i.i.d. drawn from $\mathcal{D}_t$. We use $(\hat{\mathbf{x}}, y)$ to construct a dataset $\mathcal{D}_r$ and train the perturbed model on it as [40]. Then, for $(\mathbf{x}, y) \in \mathcal{D}_e$ (the testing subset of CIFAR-10), we generate AE $\mathbf{x}'$ by four representative adversarial attacks (PGD, C&W, FAB, and Square) and utilize DFT to derive the amplitude and phase of frequency spectra of both $\mathbf{x}$ and $\mathbf{x}'$. Furthermore, images composed of adversarial amplitude/phase and benign phase/amplitude (denoted by $\mathbf{x}'_{amp}/\mathbf{x}'_{pha}$) are obtained by

$$\mathbf{x}'_{amp} = \mathcal{F}^{-1}(\mathcal{A}(\mathbf{x}'), \mathcal{P}(\mathbf{x})), \quad \mathbf{x}'_{pha} = \mathcal{F}^{-1}(\mathcal{A}(\mathbf{x}), \mathcal{P}(\mathbf{x}')).$$

For each $(\mathbf{x}, y) \in \mathcal{D}_e$, with every adopted adversarial attack, we use $(\mathbf{x}', y)$, $(\mathbf{x}'_{amp}, y)$ and $(\mathbf{x}'_{pha}, y)$ to combine evaluation datasets $\mathcal{D}_{\mathrm{AE}}$, $\mathcal{D}_{amp}$ and $\mathcal{D}_{pha}$, which are used to evaluate the robustness of

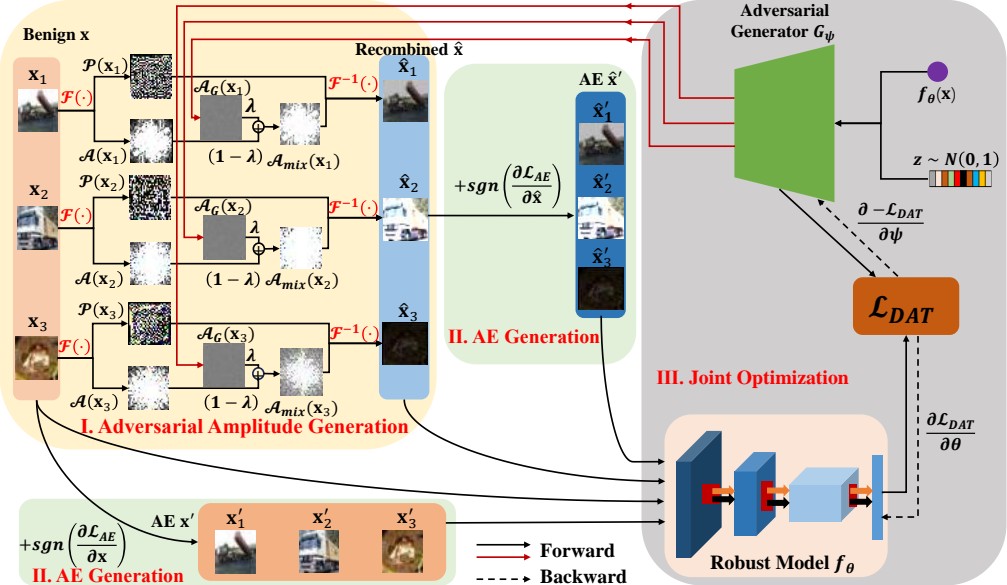

Figure 3: The overview of DAT, which consists of three stages: (I) adversarial amplitude generation, (II) AE generation, and (III) joint optimization.

the standard, robust, and perturbed models, respectively. The experimental outcomes are illustrated in Figure 2, from which we can draw the following conclusion:

**Impact of Adversarial Attacks.** As shown in Figure 2a, under all four attacks, the standard model completely cannot predict samples in $\mathcal{D}_{AE}$. Moreover, the standard model exhibits higher test accuracy on $\mathcal{D}_{amp}$ than that on $\mathcal{D}_{pha}$. These results suggest that adversarial attacks have a more substantial impact on the phase patterns than amplitude ones.

**Impact of AT.** As depicted in Figure 2b, AT simultaneously enhances the model's performance on $\mathcal{D}_{AE}$ and $\mathcal{D}_{amp}$ and $\mathcal{D}_{pha}$, in comparison to the standard model. Notably, the robust model exhibits superior performance on $\mathcal{D}_{pha}$ over $\mathcal{D}_{amp}$, contrary to the standard model. That indicates the AT helps the model learn more phase patterns unaffected by adversarial attacks, enhancing the model's robustness on both adversarial phase patterns and AEs. The phenomena indicate that by forcing the model to focus on the phase patterns of samples, the model's robustness can be improved further.

**Impact of Mixing Amplitude.** As shown in Figure 2c, for the perturbed model trained with AT on $\mathcal{D}_r$, compared with the robust one, its performance on $\mathcal{D}_{AE}$ and $\mathcal{D}_{pha}$ is improved further, while the performance on $\mathcal{D}_{amp}$ is rarely changed. From these results, we can conclude that mixing the amplitude with that of the randomly selected distractor can force the model to focus on phase patterns, learning more patterns within the phase unaffected by adversarial perturbations. Then, the model's robustness on AEs is improved further, while robustness on amplitude patterns is retained.

Following the insight from these studies, we propose DAT to enhance the model's robustness, mixing training sample's amplitude with an adversarial one, generated by the adversarially optimized AAG.

## 3 Methodology

In this section, the details of our proposed DAT are introduced and analyzed. As illustrated in Figure 3, DAT consists of three stages. It first adopts the AAG $G_\psi$ to generate adversarial amplitude and obtain the recombined data for benign ones in Stage I. With the proposed loss $\mathcal{L}_{AE}$ in Stage II, we produce the AEs for both benign and recombined samples. Then, taking both benign and recombined samples, total loss $\mathcal{L}_{DAT}$ for DAT are minimized to update robust model $f_\theta$, and maximized to optimize $G_\psi$ adversarially in Stage III. Stages I and III involve the adversarially trained AAG $G_\psi$ and commonly

updated model $f_{\boldsymbol{\theta}}$, both of which share the $\mathcal{L}_{\mathsf{DAT}}$ and optimized jointly following the objective as:

$$\min_{\boldsymbol{\theta}} \mathbb{E}_{(\mathbf{x},y)\sim\mathcal{D}} \left[ \max_{\boldsymbol{\psi}} \mathbb{E}_{\hat{\mathbf{x}}\sim p(\hat{\mathbf{x}}|\mathbf{x},\boldsymbol{\psi})} \left[ \mathcal{L}_{\mathsf{DAT}}(f_{\boldsymbol{\theta}}(\mathbf{x}), f_{\boldsymbol{\theta}}(\hat{\mathbf{x}}), y) \right] \right], \tag{3}$$

where $\hat{\mathbf{x}}$ is the recombined data of $\mathbf{x}$, following a sample-dependent conditional distribution $p(\hat{\mathbf{x}}|\mathbf{x},\boldsymbol{\psi})$. Stage II encompasses an efficient AE generation method, optimized using the proposed loss $\mathcal{L}_{\mathsf{AE}}$ as:

$$\max_{\mathbf{x}'\in\mathcal{S}_\epsilon[\mathbf{x}]} \mathbb{E}_{(\mathbf{x},y)\sim\mathcal{D}} \left[ \mathcal{L}_{\mathsf{AE}}(f_{\boldsymbol{\theta}}(\mathbf{x}), f_{\boldsymbol{\theta}}(\mathbf{x}'), y) \right]. \tag{4}$$

Following the order of these three stages, we introduce (I) AAG in Sec. 3.1 and (II) the efficient AE generation in Sec. 3.2. Subsequently, building on these components, Sec. 3.3 details (III) the joint optimization. Ultimately, Sec. 3.4 theoretically analyzes the mechanism of how the adversarial amplitude spectrum generated by AAG enforces the model to focus on the phase-level patterns.

### 3.1 Adversarial Amplitude Generator

To explain the proposed AAG, we first introduce some constraints that the recombined $\hat{\mathbf{x}}$ with perturbed amplitude is expected to meet. With a small $\epsilon_1 > 0$ and an $\epsilon_2 \gg \epsilon_1$, ideally, we expect $\hat{\mathbf{x}}$ of $(\mathbf{x}, y) \in \mathcal{D}$ based on the generated adversarial amplitude to satisfy the following three conditions:

- **C1.** $|h_p(\mathbf{x}) - h_p(\hat{\mathbf{x}})| < \epsilon_1$: Ensuring $\hat{\mathbf{x}}$ retains the same semantics in the phase spectrum as $\mathbf{x}$.

- **C2.** $F_{\boldsymbol{\theta}}(\mathbf{x}) = F_{\boldsymbol{\theta}}(\hat{\mathbf{x}})$: Ensuring $\hat{\mathbf{x}}$ remains distinguishable with the same label as $\mathbf{x}$ by $f_{\boldsymbol{\theta}}$.

- **C3.** $|h_a(\mathbf{x}) - h_a(\hat{\mathbf{x}})| > \epsilon_2$: Making $\hat{\mathbf{x}}$ maximize the $\mathcal{L}_{\mathsf{DAT}}$, causing the model's difficulty fitting the amplitude of images, and forcing the model to focus on phase patterns.

Let us first analyze the above conditions **C1-C3** when a randomly selected distractor image is used. Since $\hat{\mathbf{x}}$ is recombined by the mixed amplitude with the distractor and original phase of $\mathbf{x}$, then, **C1** can be easily satisfied. As stated in Sec. 1, when the gap between the randomly selected distractor and training sample is too large, the $\mathcal{P}(\mathbf{x})$'s information can be damaged, resulting in the inconsistent prediction between the $\mathbf{x}$ and $\hat{\mathbf{x}}$, destroying the **C1** and **C2**. Conversely, it is difficult to satisfy **C3**, limiting the model's attention to the patterns in $\mathcal{P}(\mathbf{x})$. The above statement indicates that it is difficult for $\hat{\mathbf{x}}$, using the amplitude of the randomly selected distractor, to meet the above three constraints.

Instead of searching for an appropriate distractor, we here resort to a generative approach, i.e., developing $G_{\boldsymbol{\psi}}$ to generate an adversarial amplitude $\mathcal{A}_G(\mathbf{x})$ as

$$\mathcal{A}_G(\mathbf{x}) = G_{\boldsymbol{\psi}}(\mathbf{z}, f_{\boldsymbol{\theta}}(\mathbf{x})), \quad \text{where } \mathbf{z} \stackrel{\text{i.i.d.}}{\sim} \mathcal{N}(\mathbf{0}, \mathbf{I}).$$

For efficient training, $G_{\boldsymbol{\psi}}$ is constructed by four linear layers (detailed architecture in Appendix E.2). Moreover, the input with $f_{\boldsymbol{\theta}}(\mathbf{x})$ can ease the difficulty of $G_{\boldsymbol{\psi}}$'s convergence. With $G_{\boldsymbol{\psi}}$, since $\hat{\mathbf{x}}$ is still recombined by $\mathcal{A}_G(\mathbf{x})$ and the $\mathcal{P}(\mathbf{x})$, then, **C1** is satisfied. For the outer minimization in Eq. (3), we retain the label of $\hat{\mathbf{x}}$ same as $\mathbf{x}$, minimizing $\mathcal{L}_{\mathsf{DAT}}$ to update $f_{\boldsymbol{\theta}}$, meeting the **C2**. To meets **C3**, $G_{\boldsymbol{\psi}}$ is optimized adversarially by maximizing the $\mathcal{L}_{\mathsf{DAT}}$ (further details in Sec. 3.3), shown as the inner step in Eq. (3). Thereby, $\hat{\mathbf{x}}$ can limit $f_{\boldsymbol{\theta}}$ to fit amplitude information. To achieve the convergence, $f_{\boldsymbol{\theta}}$ has to focus on patterns in $\mathcal{P}(\mathbf{x})$, learning more phase patterns unaffected by adversarial perturbations.

Since $G_{\boldsymbol{\psi}}$ is adversarially trained by maximizing the $\mathcal{L}_{\mathsf{DAT}}$, $\mathcal{A}_G(\mathbf{x})$ is likely to compromise semantic integrity [63], hindering $f_{\boldsymbol{\theta}}$ to retain the prediction consistency between $\hat{\mathbf{x}}$ and $\mathbf{x}$ [31]. Moreover, due to the loss of original amplitude information, using $\mathcal{A}_G(\mathbf{x})$ to replace $\mathcal{A}(\mathbf{x})$ entirely can also hurt the model's robustness [58]. As shown in Sec. 2, the mix-up operation on amplitude rarely has impact on the amplitude level robustness. That indicates the linear mix-up operation can preserve the energy of the amplitude spectrum and maintain the original amplitude information with less impact on the sample's original information. Otherwise, the original amplitude could be compromised, thereby hindering accurate model predictions. Following this line, a mix-up operation is employed, ensuring that a portion of the original amplitude information is preserved following:

$$\mathcal{A}_{mix}(\mathbf{x}) = \lambda \cdot \mathcal{A}_G(\mathbf{x}) + (1 - \lambda) \cdot \mathcal{A}(\mathbf{x}), \quad \text{where } \lambda \sim \mathcal{U}(0, 1).$$

The mix-up operation effectively satisfies **C1** and **C2**, ensuring $\hat{\mathbf{x}}$ remains distinguishable by $f_{\boldsymbol{\theta}}$, and keeping $\mathcal{P}(\hat{\mathbf{x}})$'s patterns closer to those in $\mathcal{P}(\mathbf{x})$ in manifold. Finally, $\hat{\mathbf{x}}$ is obtained by IDFT as

$$\hat{\mathbf{x}} = \mathcal{F}^{-1}(\mathcal{A}_{mix}(\mathbf{x}), \mathcal{P}(\mathbf{x})).$$

To elaborate on $G_{\boldsymbol{\psi}}$, we perform some experiments and provide visual results in Appendixes C and F.7. Now we can use $\hat{\mathbf{x}}$ as an augmentation of $\mathbf{x}$, introduced to Stage II for generating AEs.

## 3.2 Efficient Adversarial Example Generation

Since both $\mathbf{x}$ and $\hat{\mathbf{x}}$ are fed into Stage II for generating AE, it doubles the time consumption if we use the same AE generation strategies as existing methods, *e.g.*, PGD-AT [40] and TRADES [39]. To improve the training efficiency of DAT, we now suggest an efficient AE generation strategy. Since simply reducing the iteration step results in the difficulty of AEs' reaching the actual maximum in the $\ell_\infty$-ball [49], we propose the loss $\mathcal{L}_{\mathsf{AE}}$ to increase adversarial perturbation length in each iteration as

$$\mathcal{L}_{\mathsf{AE}}(f_{\boldsymbol{\theta}}(\mathbf{x}), f_{\boldsymbol{\theta}}(\mathbf{x}'), y) = \mathcal{L}_{\mathsf{CE}}(f_{\boldsymbol{\theta}}(\mathbf{x}'), y) + \beta \cdot \mathcal{D}_{\mathsf{KL}}(f_{\boldsymbol{\theta}}(\mathbf{x}'), f_{\boldsymbol{\theta}}(\mathbf{x})), \tag{5}$$

where $\beta$ is a weighting parameter, and $\mathcal{L}_{\mathsf{CE}}$ and $\mathcal{D}_{\mathsf{KL}}$ are cross-entropy (CE) loss and Kullback-Leibler (KL) divergence. According to [49, 50], it is effective to enlarge adversarial perturbation length for each step by maximizing the distance between $f_{\boldsymbol{\theta}}(\mathbf{x})$ and $f_{\boldsymbol{\theta}}(\mathbf{x}')$. Following this line, based on PGD-AT maximizing the $\mathcal{L}_{\mathsf{CE}}$, the proposed $\mathcal{L}_{\mathsf{AE}}$ increases the adversarial perturbation step length by maximizing the KL divergence between benign sample and its AE. Then, with $\mathcal{L}_{\mathsf{AE}}$, AEs can use fewer iterative steps to achieve the actual maximum in the $\ell_\infty$-ball. As shown by experiments in Appendix F.5, DAT only needs *5 steps* to generate AEs for both benign $\mathbf{x}$ and recombined $\mathbf{x}'$, significantly reducing the training time while maintaining the model's robustness. Enlarging the inner step size $\alpha$ (in Eq. (10) of Appendix B.3) can also increase the adversarial perturbation length in each iteration. Due to the fact that the current AT methods typically employ a fixed $\alpha$, we adopt an extra loss term for fair experimental comparisons. More details on the AE generation procedure, including pseudocodes and experimental analyses, are in Appendixes A.1, F.3 and F.5.

With the AAG and efficient AE generation, Stage III attempts to improve the model's robustness.

## 3.3 Joint Optimization

After the introduction to Stages I and II, we now delve into specifics of Stage III, joint optimization, where $f_{\boldsymbol{\theta}}$ and $G_{\boldsymbol{\psi}}$ are optimized jointly following the objective as Eq. (3). In Stage III, to satisfy **C2** in Sec. 3.1, keeping $\hat{\mathbf{x}}$ with the same label as $\mathbf{x}$ by $f_{\boldsymbol{\theta}}$, recombined and benign samples and their AEs are fed into DAT for the model training, also reducing the negative impact of amplitude information loss. Moreover, $\mathcal{L}_{\mathsf{DAT}}$ needs to be minimized on benign and recombined samples' AEs to enhance the robustness of $f_{\boldsymbol{\theta}}$, enforcing $f_{\boldsymbol{\theta}}$ to focus on the phase patterns in the meanwhile. For commonly updating $\boldsymbol{\theta}$ and adversarially renewing $\boldsymbol{\psi}$, we introduce the designed loss terms $\mathcal{L}_{\mathsf{DAT}}$ as:

$$\mathcal{L}_{\mathsf{DAT}}(f_{\boldsymbol{\theta}}(\mathbf{x}), f_{\boldsymbol{\theta}}(\hat{\mathbf{x}}), y) = \frac{1}{2}(\mathcal{L}_{\mathsf{AT}}(f_{\boldsymbol{\theta}}(\mathbf{x}), y) + \mathcal{L}_{\mathsf{AT}}(f_{\boldsymbol{\theta}}(\hat{\mathbf{x}}), y)) + \omega \cdot \mathcal{D}_{\mathsf{JS}}(f_{\boldsymbol{\theta}}(\mathbf{x}), f_{\boldsymbol{\theta}}(\hat{\mathbf{x}})), \tag{6}$$

where $\mathcal{L}_{\mathsf{AT}}$ and $\mathcal{D}_{\mathsf{JS}}$ are adversarial training and consistency regularization losses respectively, and $\omega$ is the weighting parameter for $\mathcal{D}_{\mathsf{JS}}$. We discuss $\mathcal{L}_{\mathsf{AT}}$ and $\mathcal{D}_{\mathsf{JS}}$ below separately.

**Adversarial Training Loss $\mathcal{L}_{\mathsf{AT}}$.** $\mathcal{L}_{\mathsf{AT}}$ is the loss used to guide $f_{\boldsymbol{\theta}}$ to learn robust features on AEs against adversarial attacks. As shown in Figure 1, although adversarial perturbations significantly damage phase patterns, there are still some unaffected features in the phase of AEs, important for $f_{\boldsymbol{\theta}}$ to categorize AEs correctly. Since these unaffected phase patterns contain adversarial perturbations, it is difficult for $f_{\boldsymbol{\theta}}$ to learn these features only with AEs. Therefore, we utilize benign samples and their AEs in AT, guiding the model to learn their shared features, significant for improving $f_{\boldsymbol{\theta}}$'s robustness. Following this line, $\mathcal{L}_{\mathsf{AT}}$ for $(\mathbf{x}, y) \in \mathcal{D}$ with AE $\mathbf{x}'$ can be expressed as:

$$\mathcal{L}_{\mathsf{AT}}(f_{\boldsymbol{\theta}}(\mathbf{x}), y) = \mathcal{L}_{\mathsf{CE}}(f_{\boldsymbol{\theta}}(\mathbf{x}), y) + \beta \cdot \mathcal{D}_{\mathsf{KL}}(f_{\boldsymbol{\theta}}(\mathbf{x}'), f_{\boldsymbol{\theta}}(\mathbf{x})),$$

where $\beta$ is the weighting parameter identical to that in Eq. (5). $\hat{\mathbf{x}}$ adopts the same AT loss as that of $\mathbf{x}$.

**Consistency Regularization Loss $\mathcal{D}_{\mathsf{JS}}$.** $\mathcal{D}_{\mathsf{JS}}$ is the loss used to preserve the prediction consistency between $\mathbf{x}$ and $\hat{\mathbf{x}}$. In DAT, the amplitude of $\hat{\mathbf{x}}$'s frequency spectrum is mixed with the adversarial one generated by $G_{\boldsymbol{\psi}}$, maximizing $\mathcal{L}_{\mathsf{DAT}}$, showing the large gap between $h_a(\mathbf{x})$ and $h_a(\hat{\mathbf{x}})$. Since $\hat{\mathbf{x}}$ has the same phase patterns as $\mathbf{x}$, keeping the prediction consistency between $\mathbf{x}$ and $\hat{\mathbf{x}}$ can make the model learn more phase patterns further, enhancing the model's robustness. Inspired by the mechanism, we use the Jensen–Shannon (JS) divergence $\mathcal{D}_{\mathsf{JS}}$ to ensure the prediction consistency as:

$$\mathcal{D}_{\mathsf{JS}}(f_{\boldsymbol{\theta}}(\mathbf{x}), f_{\boldsymbol{\theta}}(\hat{\mathbf{x}})) = \frac{1}{2}\left(\mathcal{D}_{\mathsf{KL}}\left(\frac{f_{\boldsymbol{\theta}}(\mathbf{x}) + f_{\boldsymbol{\theta}}(\hat{\mathbf{x}})}{2}, f_{\boldsymbol{\theta}}(\mathbf{x})\right) + \mathcal{D}_{\mathsf{KL}}\left(\frac{f_{\boldsymbol{\theta}}(\mathbf{x}) + f_{\boldsymbol{\theta}}(\hat{\mathbf{x}})}{2}, f_{\boldsymbol{\theta}}(\hat{\mathbf{x}})\right)\right). \tag{7}$$

This concludes the introduction to $\mathcal{L}_{\mathsf{DAT}}$. Notably, during the model training of DAT, the gap of batch normalization (BN) parameters between $\mathbf{x}$ and $\hat{\mathbf{x}}$ is quite large (details in Appendix D), posing challenges to model's convergence. To remedy this, we adopt different BNs for $\mathbf{x}$ and $\hat{\mathbf{x}}$ during training. Please refer to the pseudocode of DAT in Appendix A.2 for a comprehensive presentation.

### 3.4 Theoretical Analysis

Through a convergence analysis of the empirical risk, this section theoretically discusses the effects of DAT on the model to focus on phase patterns. Detailed proofs are provided in Appendix G. Concretely, we instantiate $g$ as a linear softmax classifier $\mathbf{W} = [\mathbf{w}_1, ..., \mathbf{w}_c] \in \mathbb{R}^{m \times c}$ on top of the learned features $\mathbf{h}$. Generally, for $(\mathbf{x}, y) \in \mathcal{D}$, suppose $\mathcal{T}(\mathbf{x})$ represents the augmented distribution over data points, where $\mathbf{x}$ can be transformed as anyone in $\{\mathbf{x}, \mathbf{x}', \hat{\mathbf{x}}, \hat{\mathbf{x}}'\}$. Then, the augmented data for $\mathbf{x}$ can be denoted as $t(\mathbf{x}) \sim \mathcal{T}(\mathbf{x})$. Since the augmentation will increase the discrepancy between original and augmented distributions w.h.p., we can establish a common assumption.

**Assumption 3.1.** *Assume* $\mathbb{E}_{\mathcal{T}}[\|\mathbf{h}(t(\mathbf{x})) - \mathbf{h}(\mathbf{x})\|] > \varepsilon_0$, *where* $\varepsilon_0 > 0$ *is a relatively large value. Since only the amplitude spectrum is perturbed in the proposed DAT, it is reasonable that*

$$\mathbb{E}_{\mathcal{T}}[\|h_a(t(\mathbf{x})) - h_a(\mathbf{x})\|] > \mathbb{E}_{\mathcal{T}}[\|h_p(t(\mathbf{x})) - h_p(\mathbf{x})\|].$$

**Theorem 3.2** (Weight Regularization of Amplitude Features). *Grant Assumption 3.1, when the empirical risk* $\hat{R}$ *is minimized with some convex loss function* $\mathcal{L}$ *(e.g. CE loss):*

$$\hat{R}(\mathbf{W}) \coloneqq \frac{1}{|\mathcal{D}|} \sum_{(\mathbf{x},y) \in \mathcal{D}} \mathbb{E}_{t(\mathbf{x}) \sim \mathcal{T}(\mathbf{x})} \left[ \mathcal{L}\left(\mathbf{W}^\top \mathbf{h}(t(\mathbf{x})), y\right) \right],$$

*we have* $w_{j,a} \to 0$ *for all* $j \in [c]$, *where* $w_{j,a}$ *is the corresponding weights of amplitude features* $h_a$.

**Corollary 3.3.** *Suppose the predicted probability* $f_{\mathbf{w}}(\mathbf{x}) = [p_1, \ldots, p_c]^\top$, *where*

$$p_i = \frac{\exp(\mathbf{w}_i^\top \mathbf{h})}{\sum_{j=1}^c \exp(\mathbf{w}_j^\top \mathbf{h})} = \frac{1}{\sum_{j=1}^c \exp((\mathbf{w}_j - \mathbf{w}_i)^\top \mathbf{h})}.$$

*For every* $i, j \in [c]$, *we have* $(w_{i,a} - w_{j,a})h_a \to 0$.

**Remark.** Theorem 3.2 suggests that for weights $w_{j,a}$ corresponding to features $h_a$ derived from amplitude pattern, minimizing the empirical risk $\hat{R}$ regularizes it to 0. As a result, it is difficult for the model to fit the adversarial amplitude generated by AAG. In order to converge, the model needs to reduce the reliance on $h_a$ by restricting $w_{j,a}$. Hence, the model would mitigate the impact of $h_a$ on the predicted labels, as shown in Corollary 3.3, and pay more attention to features $h_p$ derived from phase patterns, capturing more phase patterns unaffected by adversarial attacks. Therefore, we can verify the effectiveness of DAT in enhancing the model's robustness.

## 4 Experiments

In this section, we perform experiments to verify the effectiveness of DAT and explore the function of some DAT's parts. Sec. 4.1 shows experimental setups. Sec. 4.2 compares the model's robustness with existing AT methods with fixed $\epsilon$ and $\alpha$. Sec. 4.3 compares DAT with existing AT methods over complex strategy, *e.g.*, AWP and SWA. Sec. 4.4 presents the analysis of ablation studies.

### 4.1 Experimental Setup

**Experimental and Evaluation Settings.** We select three datasets: CIFAR-10, CIFAR-100, and Tiny ImageNet [17]. For all experiments in this work, ResNet-18, WideResNet-28-10 (WRN-28-10), and WideResNet-34-10 (WRN-34-10) are used as model architectures (experiments are attached to Appendix F.1), with $\beta = 15$ and $\omega = 2$ (exploration in Appendix F.4). During training, the inner step size is fixed as $\alpha = 2/255$ to generate adversarial perturbation $\ell_\infty$-bounded with constant radius $\epsilon = 8/255$ following [29, 37, 33] (details in Appendixes E.1 and E.2).

**Baselines.** We compare results with PGD-AT [40], TRADES [61], MART [54], and recent AT methods with competitive performance: LAS-AT [29], SCARL [33], and ST [37]. We also compare the DAT with OA-AT [2], DAJAT [1], and IDBH [35], which uses extra strategies, *e.g.*, AWP [56] and SWA [28]. Introduction and details of baselines are attached in Appendixes B.2 and E.3.

### 4.2 Comparison with Common Methods

In this subsection, we compare the robustness of DAT with common methods that use fixed $\epsilon$ and $\alpha$ during the training procedure. Table 1 displays the results on CIFAR-10, CIFAR-100, and

Table 1: Average natural and robust accuracy (%) of ResNet-18 against $\ell_\infty$ threat with $\epsilon = 8/255$ in 7 runs. The best results are **boldfaced**.

| DATASET | METHOD | Natural | FGSM | PGD-20 | PGD-100 | C&W$_\infty$ | AA |
|---------|--------|---------|------|--------|---------|--------------|-----|
| CIFAR-10 | PGD-AT [40] | 82.78±0.12 | 56.94±0.17 | 51.30±0.16 | 50.88±0.26 | 49.72±0.24 | 47.63±0.08 |
| | TRADES [61] | 82.41±0.12 | 58.47±0.19 | 52.76±0.08 | 52.47±0.13 | 50.43±0.17 | 49.37±0.08 |
| | MART [54] | 80.70±0.17 | 58.91±0.24 | 54.02±0.29 | 53.38±0.30 | 49.35±0.27 | 47.49±0.23 |
| | ST [37] | 83.10±0.10 | 59.42±0.32 | 54.53±0.14 | 54.31±0.17 | 51.35±0.21 | 50.51±0.17 |
| | SCARL [33] | 80.67±0.31 | 58.32±0.13 | 54.24±0.17 | 54.10±0.13 | 51.93±0.15 | 50.45±0.11 |
| | DAT (Ours) | **84.17±0.21** | **62.06±0.19** | **57.55±0.15** | **57.47±0.17** | **52.59±0.13** | **51.36±0.14** |
| | TRADES+AWP | 81.16±0.12 | 57.86±0.14 | 54.56±0.06 | 54.45±0.14 | 50.95±0.12 | 50.31±0.10 |
| | SCARL+AWP | 81.46±0.15 | 59.26±0.16 | 55.38±0.14 | 55.27±0.13 | 52.15±0.15 | 51.08±0.11 |
| | DAT+AWP (Ours) | **82.63±0.15** | **62.78±0.21** | **58.87±0.12** | **58.78±0.15** | **52.88±0.21** | **52.54±0.12** |
| CIFAR-100 | PGD-AT [40] | 57.27±0.21 | 31.81±0.11 | 28.66±0.11 | 28.49±0.16 | 26.89±0.08 | 24.60±0.04 |
| | TRADES [61] | 57.94±0.15 | 32.37±0.18 | 29.25±0.18 | 29.10±0.20 | 25.88±0.16 | 24.71±0.04 |
| | MART [54] | 55.03±0.10 | 33.12±0.26 | 30.32±0.18 | 30.20±0.17 | 26.60±0.11 | 25.13±0.15 |
| | ST [37] | 58.44±0.12 | 33.35±0.23 | 30.53±0.13 | 30.39±0.17 | 26.70±0.20 | 25.61±0.07 |
| | SCARL [33] | 57.63±0.11 | 33.14±0.19 | 30.83±0.24 | 30.77±0.21 | 26.86±0.16 | 25.82±0.19 |
| | DAT (Ours) | **62.57±0.17** | **36.63±0.12** | **33.37±0.15** | **33.15±0.12** | **28.34±0.14** | **27.11±0.15** |
| | TRADES+AWP | 58.76±0.07 | 33.82±0.15 | 31.53±0.14 | 31.42±0.12 | 27.03±0.16 | 26.06±0.12 |
| | SCARL+AWP | 58.36±0.12 | 34.25±0.14 | 32.32±0.14 | 32.26±0.13 | 27.92±0.11 | 26.83±0.15 |
| | DAT+AWP (Ours) | **63.28±0.11** | **38.22±0.14** | **35.29±0.13** | **35.18±0.12** | **29.43±0.17** | **28.09±0.12** |
| Tiny ImageNet | PGD-AT [40] | 46.36±0.22 | 23.49±0.39 | 20.41±0.29 | 20.35±0.37 | 17.86±0.28 | 14.46±0.31 |
| | TRADES [61] | 43.65±0.35 | 21.37±0.48 | 18.62±0.48 | 18.56±0.33 | 15.38±0.35 | 13.32±0.41 |
| | LAS-AT [29] | 45.27±0.35 | 24.64±0.24 | 21.82±0.27 | 21.72±0.23 | 18.07±0.25 | 16.25±0.22 |
| | SCARL [33] | 49.75±0.17 | 25.52±0.16 | 22.64±0.11 | 22.58±0.18 | 18.77±0.27 | 16.31±0.14 |
| | DAT (Ours) | **52.45±0.21** | **28.45±0.15** | **25.47±0.12** | **25.36±0.14** | **20.39±0.17** | **17.51±0.19** |
| | TRADES+AWP | 46.64±0.35 | 26.58±0.19 | 22.31±0.20 | 22.28±0.12 | 17.84±0.11 | 15.34±0.12 |
| | LAS-AT+AWP | 46.85±0.13 | 25.76±0.12 | 23.30±0.11 | 23.05±0.15 | 19.68±0.11 | 17.98±0.15 |
| | DAT+AWP (Ours) | **53.29±0.25** | **30.91±0.11** | **27.25±0.13** | **27.18±0.16** | **22.12±0.12** | **19.29±0.13** |

Table 2: Average natural and robust accuracy (%) of WRN-34-10 against $\ell_\infty$ threat with $\epsilon = 8/255$ in 7 runs. The best results are **boldfaced**.

| DATASET / METHOD | CIFAR-10 | | | | CIFAR-100 | | | |
|--------|---------|---------|--------------|-----|---------|---------|--------------|-----|
| | Natural | PGD-100 | C&W$_\infty$ | AA | Natural | PGD-100 | C&W$_\infty$ | AA |
| PGD-AT [40] | 85.37±0.74 | 54.61±0.68 | 53.42±0.82 | 52.03±0.68 | 60.63±1.17 | 30.83±0.51 | 30.21±0.83 | 27.93±0.57 |
| TRADES [61] | 85.54±0.59 | 56.04±0.45 | 53.91±0.46 | 53.37±0.51 | 61.26±0.39 | 33.11±0.42 | 30.24±0.58 | 28.32±0.62 |
| MART [54] | 85.13±0.52 | 58.72±0.66 | 53.02±0.37 | 51.61±0.48 | 60.52±0.62 | 32.34±0.62 | 29.07±0.43 | 25.91±0.36 |
| LAS-AT [29] | 86.07±0.31 | 55.97±0.47 | 55.49±0.54 | 53.34±0.42 | 61.87±0.57 | 32.21±0.45 | 30.47±0.34 | 28.91±0.39 |
| SCARL [33] | 84.41±0.23 | 57.81±0.65 | 56.21±0.47 | 54.37±0.29 | 62.41±0.36 | 34.19±0.46 | 30.53±0.31 | 29.52±0.33 |
| DAT (Ours) | **86.78±0.42** | **61.32±0.24** | **57.62±0.34** | **56.46±0.33** | **64.53±0.25** | **36.75±0.43** | **32.21±0.27** | **30.79±0.17** |

Tiny ImageNet using ResNet-18. Compared with existing methods, DAT not only improves the model's robustness but also enhances the natural accuracy. On CIFAR-10, DAT achieves an average improvement of ~2.9% against FGSM, PGD-20, and PGD-100. For challenging C&W$_\infty$ and AA, DAT obtains ~0.66% and ~0.85% improvement, respectively. Specifically, since SCARL adopts a contrastive learning strategy, DAT achieves less improvement against C&W$_\infty$ compared to it. CIFAR-100 contains more classes but fewer samples for each class, thereby making it challenging for AT. DAT enhances the model's robustness by ~2.7% on average against FGSM, PGD-20, and PGD-100 compared to the existing methods. For the demanding C&W$_\infty$ and AA, the model's robustness is improved by ~1.5% and ~1.3%. On the intricate real-world dataset Tiny ImageNet, DAT achieves ~2.9% better performance against FGSM, PGD-20, and PGD-100. For the challenging C&W$_\infty$ and AA, we enhance the model's robustness by ~1.6% and ~1.2%. AWP is proved to be an effective method for protecting the model from robust overfitting. To further improve the model's robustness, we combine DAT with AWP, and compare the performance with the combination of various existing methods and AWP fairly. Since a few methods provide the codes for the combination with AWP, we choose only these methods for comparison. As shown in Table 1, the combination of DAT and AWP improves the model's robustness further and still outperforms previous methods.

Large-capacity models usually have better adversarial robustness. Thus, we also perform comparative experiments using WRN-34-10 on CIFAR-10 and CIFAR-100, as shown in Table 2. Compared to existing methods, DAT still retains better performance and has a lower negative impact on the model's natural accuracy, achieving ~1.51%, ~0.63%, and ~1.31% robust accuracy improvement

Table 3: The average experimental results for methods with complex strategies against $\ell_\infty$ threat model with $\epsilon = 8/255$ in 7 runs. The best results are **boldfaced**.

| ARCHITECTURE | ResNet-18 | | | | WRN-34-10 | | | |
|---|---|---|---|---|---|---|---|---|
| | CIFAR-10 | | CIFAR-100 | | CIFAR-10 | | CIFAR-100 | |
| METHOD | PGD-20 | AA | PGD-20 | AA | PGD-20 | AA | PGD-20 | AA |
| TRADES+AWP | 54.56±0.06 | 50.31±0.10 | 31.53±0.14 | 26.06±0.12 | 59.26±0.24 | 55.28±0.21 | 34.48±0.26 | 29.74±0.21 |
| TRADES+AWP+SWA | 55.21±0.24 | 51.14±0.13 | 31.72±0.23 | 26.21±0.15 | 60.25±0.26 | 55.37±0.15 | 35.16±0.23 | 29.92±0.16 |
| OA-AT (SWA+variable $\epsilon$ and $\alpha$) [2] | 56.47±0.37 | 50.83±0.24 | 32.63±0.25 | 26.84±0.36 | 60.49±0.31 | 57.91±0.18 | 36.18±0.27 | 30.35±0.23 |
| DAJAT (AWP+SWA+variable $\epsilon$&$\alpha$) [1] | 56.52±0.47 | 51.85±0.26 | 32.96±0.32 | 27.83±0.29 | 62.34±0.35 | 56.62±0.23 | 37.05±0.14 | 31.51±0.17 |
| IDBH (AWP+SWA+variable $\epsilon$) [35] | 57.48±0.34 | 52.31±0.26 | 33.67±0.27 | 27.86±0.32 | 62.47±0.23 | 57.64±0.26 | 36.46±0.23 | 31.34±0.22 |
| DAT+AWP (Ours) | 58.57±0.14 | 52.54±0.12 | 35.29±0.13 | 28.09±0.12 | 63.34±0.18 | 57.96±0.16 | 38.41±0.17 | 31.62±0.12 |
| DAT+AWP+SWA (Ours) | 58.84±0.16 | 52.76±0.14 | 35.47±0.11 | 28.31±0.13 | 63.65±0.19 | 58.12±0.18 | 38.59±0.16 | 31.81±0.12 |

Table 4: Results of ablation studies with ResNet-18 against $\ell_\infty$ with $\epsilon = 8/255$ average in 7 runs.

| DATASET | CIFAR-10 | | | CIFAR-100 | | |
|---|---|---|---|---|---|---|
| METHOD | Natural | PGD-20 | AA | Natural | PGD-20 | AA |
| Baseline | 82.87±0.46 | 52.76±0.51 | 49.52±0.62 | 58.27±0.52 | 29.89±0.58 | 24.92±0.39 |
| DAT w/o $\mathcal{D}_{JS}$ | 83.76±0.49 | 57.06±0.51 | 51.25±0.22 | 60.92±0.46 | 32.83±0.47 | 26.53±0.28 |
| DAT w/o Mix-up | 81.45±0.49 | 55.64±0.54 | 50.13±0.35 | 59.84±0.42 | 30.83±0.46 | 25.41±0.29 |
| DAT w/o Split BN | 82.37±0.43 | 53.07±0.53 | 46.53±0.32 | 56.51±0.55 | 28.47±0.34 | 24.91±0.39 |
| DAT w/o AAG | 83.41±0.53 | 55.32±0.62 | 49.81±0.31 | 62.19±0.73 | 30.27±0.66 | 25.82±0.42 |
| DAT | 84.17±0.21 | 57.55±0.15 | 51.36±0.14 | 62.57±0.17 | 33.37±0.15 | 27.11±0.15 |

on CIFAR-10 with WRN-34-10. On CIFAR-100, the robustness obtained by DAT is enhanced by ~2.56%, ~1.68%, and ~1.27%, showing significant improvement compared to existing methods.

## 4.3 Comparison with Complex Strategy Based Methods

After the comparison of common methods, we then demonstrate the performance of some existing methods with extra strategies, *e.g.*, AWP and SWA. Addepalli *et al.* [1] find that variable $\epsilon$ and $\alpha$ can enhance the model's robustness, and some methods using variable $\epsilon$ and $\alpha$ (*e.g.*, DAJAT, OA-AT, and IDBH) obtain competitive performance against various adversarial attacks. To further verify the effectiveness of DAT, we compare it with the latest competitive methods, *e.g.*, DAJAT with one augmentation, OA-AT and IDBH, see Appendix B.2. As shown in Table 3, fixing $\epsilon = 8/255$ and $\alpha = 2/255$, we combine DAT with AWP and SWA. Compared with IDBH, on CIFAR-10 with ResNet-18 and WRN-34-10, DAT with AWP and SWA achieves robust improvement ~1.36% and ~1.18% against PGD-20, and ~0.45% and ~0.48% against AA. For complex CIFAR-100 with ResNet-18 and WRN-34-10, compared with the previous best method, our method obtain ~1.8% and ~1.54% improvement against PGD-20, and ~0.45% and ~0.30% enhancement against AA. Additionally, we perform with synthesized data and augmentation strategies, attached in Appendixes F.1 and F.2.

## 4.4 Ablation Study

Through experiments on CIFAR-10 and CIFAR-100, this subsection discusses the impact of different components in DAT on the model's natural accuracy and robustness against PGD-20 and AA. As shown in Table 4, we use the model with the proposed AE generation method without recombined data as the baseline. Compared with complete DAT, the robustness is decreased ~0.3% and ~0.5% on CIFAR-10 and CIFAR-100 without $\mathcal{D}_{JS}$. Due to $\mathcal{D}_{JS}$ keeping the predictions consistent between the benign and recombined data, removing $\mathcal{D}_{JS}$ makes the model learn less unaffected phase patterns and thus reduces the robust performance. As mentioned in Sec. 3.1, the mix-up operation combines the generated adversarial and original amplitude spectrum. Since the model still needs some original amplitude information, Table 4 shows that replacing the original amplitude spectrum with the generated one drops ~1.6% and ~1.9% model's robustness on CIFAR-10 and CIFAR-100, due to damaging original phase patterns. As mentioned in Sec. 3.3, the gap of BN parameters between original and recombined data is quite large. Without split BNs, the robustness of the model is reduced ~4.6% and ~3.5% on CIFAR-10 and CIFAR-100, showing that the model robustness is significantly affected because of the convergence difficulty. For experiments without AAG, we mix the amplitude of training sample's frequency spectra as Sec. 2. Table 4 shows the selected amplitude does not severely reduce the natural accuracy but significantly decreases robustness ~1.6% and ~2.2% on CIFAR-10 and CIFAR-100, confirming the effectiveness of AAG. To comprehensively

explore the effectiveness of AAG, we combine AAG with some existing AT methods on ResNet-18, see Appendixes F.6 and F.7. Additionally, we compare the time consumption in Appendix F.3.

## 5  Conclusion

This work presents a novel Dual Adversarial Training (DAT) method to improve adversarial robustness against various adversarial attacks. We first illustrate the motivation through some exploration experiments. Subsequently, we delve into the efficient and effective DAT, discussing both its underlying motivation and detailed mechanics. Additionally, we theoretically validate the functionality of the Adversarial Amplitude Generator (AAG) and the convergence properties of the DAT model. Through experiments across multiple datasets against various adversarial attacks, we verify that the proposed DAT significantly improves model robustness. We also explore the hyper-parameters and discuss the function of specific components of DAT. In the future, we will design a more suitable amplitude generation and augmentation strategy to enhance robustness further.

## Acknowledgements

This work was supported in part by Macau Science and Technology Development Fund under SKLIOTSC-2021-2023, 0072/2020/AMJ, 0022/2022/A, and 0014/2022/AFJ; in part by Research Committee at University of Macau under MYRG-GRG2023-00058-FST-UMDF and MYRG2022-00152-FST; in part by Natural Science Foundation of Guangdong Province of China under EF2023-00116-FST; in part by the Natural Science Foundation of China under Grant 62202009.

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

# Appendix

## Table of Contents

This appendix provides additional support to the main ideas presented in the submission. The general pipeline is as follows:

- §A presents the algorithms of the proposed efficient AE generation and DAT.

- §B presents comprehensive related works, including adversarial attack and adversarial training methods.

- §C presents additional details of AAG (*e.g.*, mix-up operation and $\lambda$ range).

- §D presents the gap of BN parameters for benign and recombined data, showing the reason for the split BN operation in the model training of DAT.

- §E presents more information about the training settings and selected datasets.

- §F presents omitted experimental results (*e.g.*, synthesized methods, data augmentation strategies and training time, iteration $K$ and different inputs of AAG).

- §G presents explanation and proof for Theorem 3.2.

# A Pseudocodes of AE Generation and DAT

## A.1 Pseudocode of the AE Generation Method

---
**Algorithm 1** EFFICIENT ADVERSARIAL EXAMPLE GENERATION (EAEG)
---
**Input:** Mini-batch $\mathcal{B} = \{(\mathbf{x}_i, y_i)\}_{i=1}^M$, model $f_{\boldsymbol{\theta}}$, perturbation radius $\epsilon$, iteration step $K$, step size $\alpha$, weight parameter $\beta$
**Output:** AEs $\{\mathbf{x}_i'\}_{i=1}^M$

    // EAEG Procedure
1: **for** $i = 1, \ldots, M$ (in parallel) **do**
2:     $\mathbf{x}_i' \leftarrow \mathbf{x}_i + 0.001 \cdot \boldsymbol{\xi}$, where $\boldsymbol{\xi} \overset{\text{i.i.d.}}{\sim} \mathcal{N}(\mathbf{0}, \mathbf{I})$          ▷ *initialize AE*
3:     **for** $k = 1$ to $K$ **do**
4:        $\mathcal{L}_{\mathsf{AE}}(\mathbf{x}_i'; \mathbf{x}_i, y_i, \boldsymbol{\theta}) = \mathcal{L}_{\mathsf{CE}}(f_{\boldsymbol{\theta}}(\mathbf{x}_i'), y_i) + \beta \cdot \mathcal{D}_{\mathsf{KL}}(f_{\boldsymbol{\theta}}(\mathbf{x}_i'), f_{\boldsymbol{\theta}}(\mathbf{x}_i))$      ▷ *Eq.* (5)
5:        $\mathbf{x}_i' \leftarrow \Pi_{\mathcal{S}_\epsilon[\mathbf{x}_i']} \left( \mathbf{x}_i' + \alpha \cdot \text{sign} \left( \nabla_{\mathbf{x}_i'} \mathcal{L}_{\mathsf{AE}}(\mathbf{x}_i'; \mathbf{x}_i, y_i, \boldsymbol{\theta}) \right) \right)$      ▷ *update AE*
---

## A.2 Pseudocode of DAT

---
**Algorithm 2** DUAL ADVERSARIAL TRAINING (DAT)
---
**Input:** Training data $\mathcal{D} = \{(\mathbf{x}_i, y_i)\}_{i=1}^N$, mini-batch size $M$, model $f_{\boldsymbol{\theta}}$, Adversarial Amplitude Generator $G_{\boldsymbol{\psi}}$, AE generation method EAEG, perturbation radius $\epsilon$, iteration step $K$, step size $\alpha$, weight parameter $\{\beta, \omega\}$, training epoch $T$
**Output:** Robust model $f_{\boldsymbol{\theta}}$

    // DAT Procedure
1: **for** $t = 1$ to $T$ **do**
2:     **for** each batch $\mathcal{B} = \{(\mathbf{x}_i, y_i)\}_{i=1}^M$ **do**
3:        **for** $i = 1, \ldots, M$ (in parallel) **do**
4:           $\mathbf{x}_i' = \text{EAEG}(\mathbf{x}_i, y_i; \boldsymbol{\theta}, \epsilon, K, \alpha, \beta)$      ▷ *generate AE for benign data*
5:           $\mathcal{A}_G(\mathbf{x}_i) = G_{\boldsymbol{\psi}}(\mathbf{z}, f_{\boldsymbol{\theta}}(\mathbf{x}_i))$, where $\mathbf{z} \overset{\text{i.i.d.}}{\sim} \mathcal{N}(\mathbf{0}, \mathbf{I})$      ▷ *generate adversarial amplitude*
6:           $\mathcal{A}_{mix}(\mathbf{x}_i) = \lambda \mathcal{A}_G(\mathbf{x}_i) + (1 - \lambda)\mathcal{A}(\mathcal{F}(\mathbf{x}_i))$, where $\lambda \sim \text{Uniform}(0, 1)$      ▷ *mix amplitudes*
7:           $\hat{\mathbf{x}}_i = \mathcal{F}^{-1}(\mathcal{A}_{mix}, \mathcal{P}(\mathcal{F}(\mathbf{x}_i)))$      ▷ *generate recombined sample*
8:           $\hat{\mathbf{x}}_i' = \text{EAEG}(\hat{\mathbf{x}}_i, y_i; \boldsymbol{\theta}, \epsilon, K, \alpha, \beta)$      ▷ *generate AE for recombined sample*
9:           $\mathcal{L}_{\mathsf{DAT}}(f_{\boldsymbol{\theta}}(\mathbf{x}_i), y_i) = \frac{1}{2}\left(\mathcal{L}_{\mathsf{AT}}(f_{\boldsymbol{\theta}}(\mathbf{x}_i), y_i) + \mathcal{L}_{\mathsf{AT}}(f_{\boldsymbol{\theta}}(\hat{\mathbf{x}}_i), y_i)\right) + \omega \cdot \mathcal{D}_{\mathsf{JS}}(f_{\boldsymbol{\theta}}(\hat{\mathbf{x}}_i), f_{\boldsymbol{\theta}}(\mathbf{x}_i))$
                                                                                   ▷ *Eq.* (6)
10:        Update $\boldsymbol{\psi}$ and $\boldsymbol{\theta}$ by $\mathcal{L}_{\mathsf{DAT}}$
---

# B  Related Work & Background

## B.1  Adversarial Attacks

With the heightened attention on DNN vulnerabilities, numerous adversarial attack methods have developed to study the model's robustness. FGSM [22] generates AEs by applying a gradient on benign samples in a single iteration step. Building upon I-FGSM [34], PGD [40] emerges as the strongest first-order attack, amplifying the adversarial impact of AEs. In order to reduce the difficulty of parameter settings in FGSM, the accurate and efficient Deepfool [42] is developed. To enhance the effect of PGD attack, APGD, and APGD-DLR are proposed [13]. These methods have alternative loss functions and do not require an inner step size in the AEs' generation procedure. However, these white-box attacks require detailed knowledge of the target model to generate AEs, which poses challenges in practical scenarios. To address this challenge, black-box [45, 9, 27, 10, 57, 24] and no-box [45, 36] adversarial attacks are proposed. For a comprehensive evaluation of model robustness, ensemble adversarial attacks consisting of multiple types of adversarial attacks become popular. AutoAttack (AA) [13] stands out as a representative ensemble attack approach, seamlessly integrating white-box attacks including APGD, APGD-DLR, and FAB [12], alongside a black-box attack called Square attack [3]. AA is broadly recognized as a benchmark tool for the robustness evaluation of models.

## B.2  Adversarial Training

In an effort to protect models from adversarial attacks, various methodologies have been developed. Adversarial training is one of the most effective methods for countering adversarial attacks [64, 62, 4, 5, 19, 20, 40, 61, 1, 35]. PGD-AT [40] formulates AT as a min-max optimization problem. AEs are first generated via PGD and then fed into the trained model to minimize empirical risk. To address the robust overfitting issues, early stopping of PGD-AT is proposed to further enhance the model's robustness [48]. However, PGD-AT only takes AEs into the training procedure. This approach, due to the data distribution shift between AEs and benign samples, can diminish the model's accuracy. To make a better tradeoff between accuracy and robustness, TRADES [61] takes both benign samples and AEs into the AT procedure. Motivated by weight perturbation methods in standard training, Adversarial Weight Perturbation (AWP) [56] adversarially perturbs both inputs and weights, markedly easing the robust overfitting issues and improving adversarial robustness. Additionally, Stochastic Weight Averaging (SWA) [28] is also a frequently used technology to reduce the negative impact of robust overfitting issues and enhance the model's robustness. Misclassification Aware adveRsarial Training (MART) [54] adopts misclassified training samples as regularizers to enhance the model's robustness against AEs generated by various adversarial attacks. Instead of generating AEs by maximizing the prediction distance between AEs and benign samples as in TRADES, Squeeze Training (ST) [37] selects a better reference target and uses collaborative examples to benign ones, produced by minimizing the prediction distance between collaborative and benign samples. Different from existing hand-crafted strategy based AT, Adversarial Training with Learnable Attack Strategy (LAS-AT) [29] generates AEs by a reinforcement learning network. Motivated by Contrastive Language-Image Pre-training (CLIP) [46], Semantic Constraint Adversarial Robust Learning (SCARL) [33] uses the text information to obtain robust semantic information of training samples, improving the model's robustness. Due to the positive impact of data augmentation on AT, Diverse Augmentation-based Joint Adversarial Training (DAJAT) [1] uses AutoAugment, SWA, and AWP to achieve an effective AT. To reduce the time consumption of AE generation, DAJAT adopts variable adversarial perturbation radius $\epsilon$ and inner step size $\alpha$ to reduce the iteration steps. OA-AT [2] uses SWA and variable $\epsilon$ and $\alpha$ to protect the model from AEs with large adversarial magnitudes. Li *et al.* [35] show that the effectiveness of data augmentations on robustness improvement depends on their impacts on the model's robust accuracy on test data, referred to as hardness. Moreover, the work proves that augmentation strategies with moderate hardness can protect the model from robust overfitting issues and enhance the model's robustness. Based on the phenomena, with variable adversarial perturbation radius $\epsilon$, Improved Diversity and Balanced Hardness (IDBH) [35] combines several data augmentation strategies to obtain significant robustness improvement against various adversarial attacks. Recently, motivated by the stable diffusion (SD) model [30], some methods [47, 23, 15, 55, 44] synthesize data by SD to enhance the model's robustness further.

### B.3 Generation of Adversarial Example in Adversarial Training

PGD-AT [40] formulates a min-max strategy to inject AEs into AT. The optimization objective is

$$\min_{\boldsymbol{\theta}} \sum_{i=1}^{N} \max_{\mathbf{x}'_i \in \mathcal{S}_\epsilon[\mathbf{x}_i]} \mathcal{L}_{\mathsf{CE}}(f_{\boldsymbol{\theta}}(\mathbf{x}'_i), y_i), \tag{8}$$

where $\mathbf{x}'_i$ is the AE for $\mathbf{x}_i$, and $\mathcal{L}_{\mathsf{CE}}$ represents the cross-entropy (CE) loss. To achieve a better tradeoff between natural and robust accuracy, TRADES [61] incorporates both AEs and benign samples into training as

$$\min_{\boldsymbol{\theta}} \sum_{i=1}^{N} \left( \mathcal{L}_{\mathsf{CE}}(f_{\boldsymbol{\theta}}(\mathbf{x}_i), y_i)) + \beta \cdot \max_{\mathbf{x}'_i \in \mathcal{S}_\epsilon[\mathbf{x}_i]} \mathcal{D}_{\mathsf{KL}}(f_{\boldsymbol{\theta}}(\mathbf{x}'_i), f_{\boldsymbol{\theta}}(\mathbf{x}_i)) \right), \tag{9}$$

where $\beta$ is a weight parameter, and $\mathcal{D}_{\mathsf{KL}}$ represents the Kullback-Leibler (KL) divergence.

For $(\mathbf{x}, y) \in \mathcal{D}$, its corresponding AE $\mathbf{x}'$ is initialized by adding a random noise (*e.g.*, uniform noise in PGD and Gaussian noise in TRADES) on $\mathbf{x}$, then iteratively updated by $K$ steps following

$$\mathbf{x}'^{(k+1)} = \Pi_{\mathcal{S}_\epsilon[\mathbf{x}]} \big( \mathbf{x}'^{(k)} + \alpha \cdot \mathrm{sign}(\nabla_{\mathbf{x}'^{(k)}} \mathcal{L}(f_{\boldsymbol{\theta}}(\mathbf{x}'^{(k)}), y)) \big), \tag{10}$$

where $\Pi(\cdot)$ is the projection operator, $\alpha$ is the projection inner step size, and $k \in \{0, ..., K-1\}$, with $K$ typically set to 10.

### B.4 Discrete Fourier Transform

DFT transforms an image signal from the spatial domain into the frequency domain, while the inverse discrete Fourier transform (IDFT) reverses this process. Let $\mathcal{F}(\cdot)$ and $\mathcal{F}^{-1}(\cdot, \cdot)$ denote the DFT and IDFT functions, respectively. Typically, DFT is independently applied to each channel of an image within the pixel space. An image $\mathbf{x}$ can be transformed into the frequency domain as follows:

$$\mathcal{F}(\mathbf{x})(u, v) = \sum_{h=1}^{H} \sum_{w=1}^{W} \mathbf{x}(h, w) \, e^{-i2\pi(u\frac{h}{H} + v\frac{w}{W})},$$

where $(h, w)$ denotes the pixel coordinates of $\mathbf{x}$, and $(u, v) \in [H] \times [W]$ signifies coordinates in the frequency domain. The real and imaginary parts of $\mathcal{F}(\mathbf{x})$ are denoted by $\mathrm{Re}(\mathcal{F}(\mathbf{x}))$ and $\mathrm{Im}(\mathcal{F}(\mathbf{x}))$, respectively. Then, the amplitude spectrum $\mathcal{A}(\mathbf{x})$ and phase spectrum $\mathcal{P}(\mathbf{x})$ are defined as follows:

$$\mathcal{A}(\mathbf{x}) = \left( \mathrm{Re}^2(\mathcal{F}(\mathbf{x})) + \mathrm{Im}^2(\mathcal{F}(\mathbf{x})) \right)^{\frac{1}{2}}, \quad \mathcal{P}(\mathbf{x}) = \arctan \left( \frac{\mathrm{Im}(\mathcal{F}(\mathbf{x}))}{\mathrm{Re}(\mathcal{F}(\mathbf{x}))} \right). \tag{11}$$

## C Experiments of the Amplitude Spectrum Operation

To better explore the proposed AAG, we present the amplitude spectrum mix-up operation visually and explore the impact of the $\lambda$ range on the model's robustness in this part. First, we display and explain the visual results of the recombined samples and show the impact of the mix-up operation on the model's performance. Then, the experimental results of DAT with different ranges of $\lambda$ are shown.

### C.1 Visualization of the Mix-up Operation on Amplitude Spectrum

Figure 8 (see page 28) shows the benign samples, amplitude spectrum, phase spectrum, generated amplitude spectrum, and recombined data. Compared to images with mixed amplitude spectrum, data recombined by replacing the original amplitude from the generated one entirely has a great gap for the original benign samples, reducing the natural and robust accuracy of the benign test samples. To better show the impact of the different strategies on the model's performance, we perform experiments with and without the mix-up operation. Table 5 shows the DAT with mix-up has a better performance. That indicates the model still needs some original information on the amplitude spectrum, showing the effectiveness of the mix-up operation.

Table 5: Average experimental results of DAT with and without Mix-up on CIFAR-10 and CIFAR-100 with ResNet-18 and $\epsilon = 8/255$ in 7 runs.

| METHOD | CIFAR-10 | | | CIFAR-100 | | |
|---|---|---|---|---|---|---|
| | Natural | PGD-20 | AA | Natural | PGD-20 | AA |
| DAT w/o Mix-up | 81.45±0.49 | 55.64±0.54 | 50.13±0.35 | 59.84±0.42 | 30.83±0.46 | 25.41±0.29 |
| DAT w/ Mix-up | **84.17±0.21** | **57.55±0.15** | **51.36±0.14** | **62.57±0.17** | **33.37±0.15** | **27.11±0.15** |

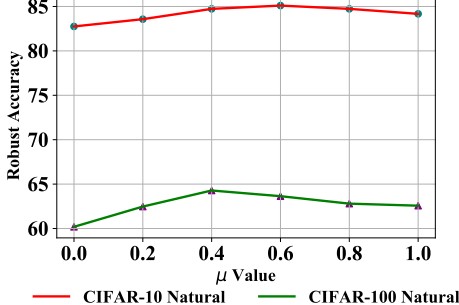

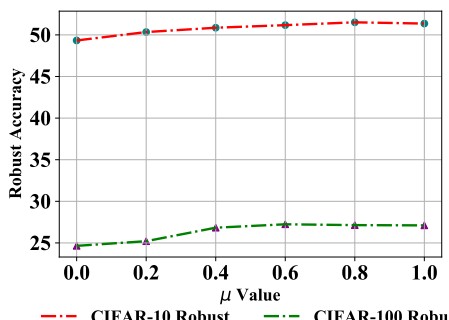

(a) Natural accuracy of DAT with different $\mu$.

(b) Robust accuracy against AA of DAT with different $\mu$.

Figure 4: Natural and robust accuracy (%) against AA of DAT with different $\mu$ on CIFAR-10 and CIFAR-100 with ResNet-18.

## C.2 Impact of the Range of Amplitude Mixture Parameters

In this subsection, to fully explore the impact of the $\lambda$ range on the model's robustness, we perform some experiments to show the impact of different $\lambda$ on the natural and robust accuracy. As shown in the work, the mix-up operation to perturb $\mathcal{A}(\mathbf{x})$ by the generated $\mathcal{A}_G(\mathbf{x})$ follows

$$\mathcal{A}_{mix}(\mathbf{x}) = \lambda \mathcal{A}_G(\mathbf{x}) + (1 - \lambda)\mathcal{A}(\mathbf{x}), \tag{12}$$

where $0 < \lambda < 1$, reserving some information of $\mathcal{A}(x)$. Suppose that $\lambda \sim \text{Uniform}(0, \mu)$, where $0 < \mu < 1$, we perform experiments with different values of $\mu$ on CIFAR-10 and CIFAR-100 against AA, where $\beta = 15$ and $\omega = 2$. As shown in Figures 4a and 4b, the test accuracy fluctuations between different values of $\mu$ are quite large, and the best settings of $\mu$ for natural and robust accuracy for CIFAR-10 and CIFAR-100 are different. For CIFAR-10, the model achieves the best natural performance when $\mu = 0.6$, while its robust accuracy is better with $\mu = 0.8$. For results on CIFAR-100, $\mu = 0.4$ makes the best natural accuracy when the model achieves better robustness with $\mu = 0.6$. In the work, there are already two hyper-parameters: $\omega$ and $\beta$. Moreover, the range of $\lambda$ is more likely to influence the settings of $\omega$ and $\beta$. Consequently, we do not discuss the impact of different ranges of $\lambda$ on the model's robust and natural accuracy in the main paper.

For the real-valued $\mathbf{x}$, it is noticed that $\mathcal{F}(\cdot)$ need to be even-conjugate, i.e., $\mathcal{F}(\mathbf{x})(-u, -v) = \overline{\mathcal{F}(\mathbf{x})(u, v)}$, implying that the amplitude spectrum is symmetric. Conversely, IDFT returns real-valued signals for a symmetric amplitude spectrum. Consequently, $G_\psi$ only generates $\mathcal{A}_G$ with the non-redundant and non-negative part of the amplitude spectrum.

## D  Batch Normalization Parameter Analysis

In the DAT training procedure, we use different BNs for the benign data and their recombined samples. In the ablation study, we show the model performances with and without a split BN are quite different, showing that the single BN significantly influences the model's performance. To better present the necessity of the split BN in DAT training, we show the cosine similarity of the parameters of each BN layer between benign data and recombined ones. There are 20 BN layers in ResNet-18. From Figures 5a and 5b, on both CIFAR-10 and CIFAR-100, we can see the cosine similarity gaps for BN parameters in different layers. The distance between the mean and variance of each BN layer is small, while the gaps between the $\gamma$s and $\beta$s are large, especially in the low-level layers and the

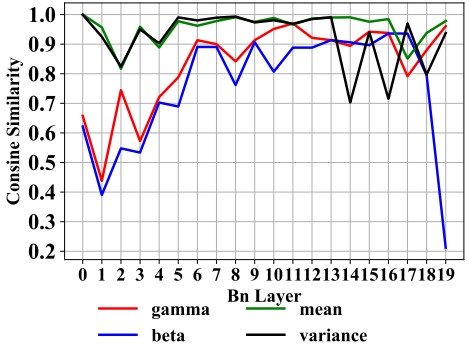 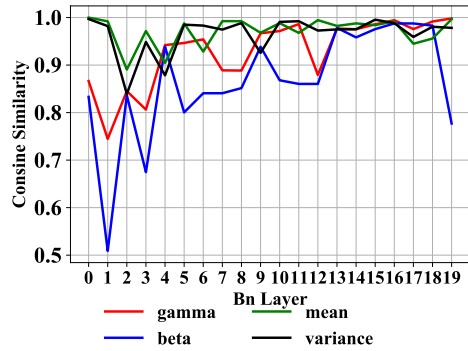

(a) Cosine similarity between the BN parameters on CIFAR-10.

(b) Cosine similarity between the BN parameters on CIFAR-100.

Figure 5: Cosine similarity between the BN parameters of the original data and recombined ones in ResNet-18 on (a) CIFAR-10 and (b) CIFAR-100.

last layer. Consequently, using a single BN in the training procedure of DAT is likely to result in the difficulty of model convergence, showing as the performance comparisons in the ablation study. To solve the issue, we adopt different BNs for the original data and the recombined ones. In the model training procedure of DAT, we split the BN into two groups: BN-A and BN-B, for each layer with BN, original data and its AE are processed by BN-A, while recombined one and its AE are regularized by BN-B.

# E  Detailed Experimental Setup

In this section, we add more details about the datasets and experimental settings. Moreover, the information about the robust evaluation is further introduced.

## E.1  Datasets

To evaluate the performance of DAT, we select three widely used benchmark datasets for robust evaluation: CIFAR-10, CIFAR-100, and complex Tiny ImageNet [17]. CIFAR-10 and CIFAR-100 contain 50,000 32×32 training samples and 10,000 32×32 test images, categorized into 10 and 100 classes respectively. Tiny ImageNet is a challenge 200-class real-world dataset, where there are 500 training and 50 test images for each category, where the image size is 64×64. Moreover, due to the samples in the test set of Tiny ImageNet without labels, we evaluate the robustness of the validation set following [29, 33].

## E.2  Training Settings

We use ResNet-18, WRN-34-10, and WRN-28-10 as model architectures. Experiments with ResNet-18 are performed on Ubuntu 20.04.3 LTS GPU server with Intel Xeon 5120 and 5×3090 by PyTorch 2.0, while WRN-34-10 and WRN-28-10 experiments are performed on DGX with a H800 GPU on PyTorch 2.0. In the model training procedure, we adopt an SGD optimizer with momentum 0.9 and weight decay 5e-4. For the common experiments, the model is trained for 150 epochs for CIFAR-10 and CIFAR-100 and 100 epochs for Tiny ImageNet. Moreover, the learning rate follows the schedule $[0.1, 0.01, 0.001]$ in decay epoch schedule $[100, 110]$ in CIFAR-10 and CIFAR-100 and in decay epoch schedule $[75, 80]$ for Tiny ImageNet. For these datasets, the experiments with AWP or the combination of AWP and SWA use the same learning rate schedule with decay epoch as $[100, 150]$.

For the proposed AAG, we use a four-linear layer structure to build it. Since only the non-negative part of the amplitude spectrum is produced, the output of AAG is processed by a Sigmoid function. In the model training procedure, we set the dimension of $\mathbf{z}$ as $\tau = 100$. Due to the input of AAG combined by $\mathbf{z}$ and sample logits extracted by $f_{\boldsymbol{\theta}}$, the input dimension of AAG is the sum of $\tau$ and class number $c$ of $\mathcal{D}$.

The AAG is optimized by an SGD optimizer with a fixed learning rate 0.1, momentum 0.9, and weight decay 5e-4. For hyper-parameters, $\beta$ and JS weight parameter $\omega$ are set as 15 and 2 respectively. During the training procedure, we adopt the basic data augmentation strategies, Random Crop and Random Horizontal Flip, for all selected datasets. For AE generation, the inner step size $\alpha$ is set to $^2/_{255}$ with $K = 5$ to generate adversarial perturbation $\ell_\infty$-bounded with radius $\epsilon = ^8/_{255}$ following previous work [29, 37, 33].

### E.3 Baselines

For experimental result comparisons in this work, we select two typical PGD-AT and TRADES [40, 61] as baselines. Moreover, we adopt two types of existing methods to perform experimental result comparisons. For common type AT methods, which are trained with fixed $\epsilon = ^8/_{255}$ and inner step size $\alpha = ^2/_{255}$ and without any extra technologies, we select MART [54], ST [37], LAS-AT [29] and SCARL [33]. Additionally, some methods with complex strategies, such as AWP, SWA, variable $\epsilon$, and changeable $\alpha$ are also selected as baselines. For AutoAugment-based DAJAT [1] consisting of AT, SWA, variable $\epsilon$ and $\alpha$, we select it with one augmentation for a fair comparison. OA-AT [2] is a variable $\epsilon$ based AT method with SWA. IDBH [35] is an AT method based on the augmentation combination of Cropping, CutOut, and ColorShape, and is combined with AWP and SWA strategies.

### E.4 Robustness Evaluation

For the experimental results, we report the model's natural and robust accuracy. We select several widely used types of adversarial attacks to generate AEs for robustness evaluation. FGSM, PGD-20, PGD-100, and C&W$_\infty$ are selected as basic methods to evaluate the model's robustness. To better show the model's robust generalization against different adversarial attacks, we also select AA consisting of black-box and white-box methods to evaluate the robustness of the model. The adversarial perturbation of these methods is $\ell_\infty$-bounded with radius $\epsilon = ^8/_{255}$ and inner step size $\alpha = ^2/_{255}$.

## F    Additional Results and Discussion

Due to the limitation of the pages, we perform some experiments on CIFAR-10 and CIFAR-100 to show the effectiveness of our proposed DAT further. In the first section, we show the experimental results on generated data with WRN-28-10. The experimental comparison between proposed DAT and some data augmentation strategies are presented in the second section. Then, we show the time consumption comparison between DAT and existing methods in the third part. The fourth subsection discusses the impact of different $\beta$ and $\omega$. Then, the impact of iteration $K$ of AE generation on the model's performance is explored in the fifth subsection. In the sixth part, the AAG is combined with existing methods to verify its effectiveness. Then, we explore the influence of AAG's input. The last two subsections show the results comparison of single and dual AE generation of DAT and limitations of the work.

### F.1    Comparison for Different AT Methods with Generated Data

In this subsection, following [47, 23, 15, 55, 44], we take advantage of the diffusion model to synthesize data enhance the model's robustness further. Since these existing methods are combined with strategies such as AWP, SWA, variable $\epsilon$ and changed $\alpha$, we use the DAT with AWP and SWA to show the experimental results comparison. Due to methods only providing single-time results, we just use the one-time test accuracy of DAT to compare with them on WRN-28-10. From Table 6, compared with IKL-AT [15], we can see DAT achieves about $\sim$0.69% and $\sim$0.48% robustness improvement with 1M and 20M generated data on CIFAR-10. On CIFAR-100, as shown in Table 7, for current IKL-AT, the method with the best robustness against AA with WRN-28-10 on the RobustBench [11], the model robustness is enhanced by $\sim$0.29% and $\sim$0.23% with 1M and 50M synthesized samples, respectively.

Table 6: The average experimental results for different augmentations against $\ell_\infty$ threat model with $\epsilon = 8/255$ on CIFAR-10. #Aug. refers to the number of augmentation. The best results are **boldfaced**.

| METHOD | Architecture | #Aug. | Natural | AA |
|---|---|---|---|---|
| Rebuffi *et al.* [47] | WRN-28-10 | 1M | 87.33 | 60.73 |
| Gowal *et al.* [23] | WRN-28-10 | 100M | 87.50 | 63.38 |
| Wang *et al.* [55] | WRN-28-10 | 1M | 91.12 | 63.35 |
| Wang *et al.* [55] | WRN-28-10 | 50M | 92.27 | 67.17 |
| Wang *et al.* [55] | WRN-28-10 | 20M | 92.44 | 67.31 |
| IKL-AT [15] | WRN-28-10 | 1M | 90.75 | 63.54 |
| IKL-AT [15] | WRN-28-10 | 20M | 92.16 | 67.75 |
| DAT+AWP+SWA (Ours) | WRN-28-10 | 1M | 91.37 | 64.25 |
| DAT+AWP+SWA (Ours) | WRN-28-10 | 20M | 92.86 | 68.18 |

Table 7: The average experimental results for different augmentations against $\ell_\infty$ threat model with $\epsilon = 8/255$ on CIFAR-100. #Aug. refers to the number of augmentation. The best results are **boldfaced**.

| METHOD | Architecture | #Aug. | Natural | AA |
|---|---|---|---|---|
| Pang *et al.* [44] | WRN-28-10 | 1M | 62.08 | 31.40 |
| Rebuffi *et al.* [47] | WRN-28-10 | 1M | 62.41 | 32.06 |
| Wang *et al.* [55] | WRN-28-10 | 1M | 68.06 | 35.65 |
| Wang *et al.* [55] | WRN-28-10 | 50M | 72.58 | 38.83 |
| IKL-AT [15] | WRN-28-10 | 1M | 68.99 | 35.89 |
| IKL-AT [15] | WRN-28-10 | 50M | 73.85 | 39.18 |
| DAT+AWP+SWA (Ours) | WRN-28-10 | 1M | 69.52 | 36.18 |
| DAT+AWP+SWA (Ours) | WRN-28-10 | 50M | 74.86 | 39.41 |

## F.2 Comparison of DAT with Different Augmentations

Since the recombined data based on AAG can be regarded as an augmentation for benign samples, to better present the effectiveness of the proposed strategy, we combine the proposed AE generation method with three widely used augmentation policies in AT, such as CutOut [18], CutMix [60], and AutoAugment [14], and perform experiments on CIFAR-10 and CIFAR-100. The settings of CutMix and CutOut are as [35], while AutoAugment follows [1]. In these experiments, the baseline is AT with the proposed AE generation strategy. These selected data augmentation strategies adopt the same training strategy as DAT. As shown in Table 8, for the baseline, these augmentations enhance the model's robustness against PGD-20 and AA. Compared to other strategies, DAT achieves robustness improvement of ∼1% on CIFAR-10 and ∼1.2% on CIFAR-100.

## F.3 Comparison of Training Time Consumption

In this part, we compare the time consumption of different methods on CIFAR-10 and CIFAR-100. From Table 9, we can obtain the time consumption of DAT is slightly higher than PGD-AT and TRADES because of added AAG. Compared with recently proposed methods, DAT takes less time and achieves better robustness.

Table 8: The average experimental results for different augmentations against $\ell_\infty$ threat model with $\epsilon = 8/255$ in 7 runs. The best results are **boldfaced**.

| METHOD | CIFAR-10 | | CIFAR-100 | |
|---|---|---|---|---|
| | PGD-20 | AA | PGD-20 | AA |
| Baseline | 53.13±0.51 | 49.64±0.62 | 30.09±0.58 | 25.43±0.39 |
| CutOut [18] | 55.85±0.51 | 50.28±0.14 | 31.35±0.44 | 26.26±0.14 |
| CutMix [60] | 55.76±0.42 | 50.13±0.54 | 31.26±0.62 | 26.17±0.19 |
| AutoAugment [14] | 56.24±0.45 | 50.42±0.15 | 31.69±0.52 | 26.44±0.17 |
| DAT (Ours) | **57.55±0.15** | **51.36±0.14** | **33.37±0.15** | **27.11±0.15** |

Table 9: Time consumption (s) of each training epoch for different AT methods.

| METHOD | CIFAR-10 | CIFAR-100 |
|---|---|---|
| PGD-AT [40] | 187 | 188 |
| TRADES [61] | 187 | 192 |
| ST [37] | 320 | 326 |
| SCARL [33] | 221 | 228 |
| DAT (Ours) | 218 | 221 |

## F.4 Hyper-parameter Exploration

After the performance comparisons, we then explore the settings of $\beta$ and $\omega$ by experiments on CIFAR-10 and CIFAR-100. Additionally, the $\lambda$'s range and the iteration step $K$ of AE generation are also discussed, attached to Appendixes C.2 and F.5.

**Setting of $\beta$.** As shown in Figure 6a of Appendix F.4, we present the model's natural and robust accuracy with different $\beta$ with $\omega = 2$ and $K = 5$. We can see that the curves of robust accuracy rise first and then fall. Meanwhile, natural accuracy keeps decreasing as $\beta$ increases, indicating that $\beta$ significantly influences the model's natural and robust accuracy. To achieve a better tradeoff between robust and natural accuracy, we set $\beta = 15$ for experiments of DAT on all datasets.

**Setting of $\omega$.** With $\beta = 15$ and $K = 5$, we discuss the settings of $\omega$. Figure 6b shows the model's accuracy changes along with $\omega$ on both natural and AE. When $\omega < 2$, the curves of robust accuracy rise. On the contrary, $\omega > 2$ results in the performance decreasing. Unlike the robust accuracy, the model's performance on benign samples grows with the increase of $\omega$. For better model robustness, $\omega$ is set as 2 in this work.

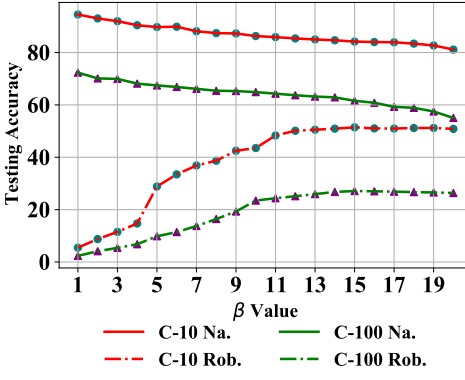
(a) Impact of $\beta$ on natural and robust accuracy.

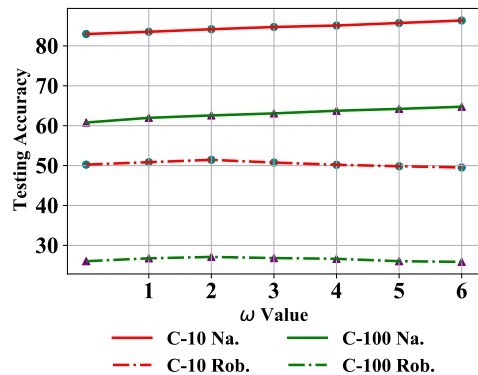
(b) Impact of $\omega$ on natural and robust accuracy.

Figure 6: The impact of $\omega$ and $\beta$ on the model's natural (Na.) and robust accuracy (Rob.) against AA on CIFAR-10 (C-10) and CIFAR-100 (C-100) with ResNet-18.

## F.5 Iteration Step $K$ of AE Generation

The generation iteration step $K$ has an extremely important impact on the model's robust accuracy. As shown in Figure 7, the larger the iteration step, the adversarial robustness against AA of the model rises with the increase of $K$. However, the larger $K$ will significantly increase the time consumption of AT. To achieve a tradeoff between the model's robustness and time consumption, we set the iteration step as $K = 5$. Then, the time consumption of DAT will not be increased compared with existing methods.

## F.6 AAG with Existing AT Methods

To further show the effectiveness of the proposed AAG, we combine AAG with PGD-AT and TRADES and perform experiments on CIFAR-10 and CIFAR-100 against PGD-20 and AA. Compared

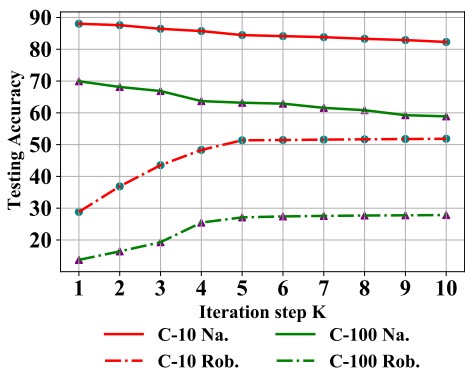

Figure 7: The impact of different iteration step $K$ on the model's performance on natural data and against AA on CIFAR-10 and CIFAR-100 with ResNet-18.

with PGD-AT and TRADES, the combinations with AAG enhance the model's robust and natural accuracy further.

Table 10: Average experimental results of AAG with existing AT methods on CIFAR-10/100 with ResNet-18 and $\epsilon = 8/255$ in 7 runs.

| DATASET | CIFAR-10 | | | CIFAR-100 | | |
|---|---|---|---|---|---|---|
| METHOD | Natural | PGD-20 | AA | Natural | PGD-20 | AA |
| PGD-AT | 82.78±0.12 | 51.30±0.16 | 47.63±0.08 | 57.27±0.21 | 28.66±0.11 | 24.60±0.04 |
| PGD-AT+AAG | **83.24±0.21** | **55.64±0.18** | **50.96±0.14** | **59.34±0.23** | **31.82±0.25** | **26.16±0.16** |
| TRADES | 82.41±0.12 | 52.76±0.19 | 49.37±0.08 | 57.94±0.15 | 29.25±0.18 | 24.71±0.04 |
| TRADES+AAG | **84.36±0.38** | **55.45±0.42** | **50.75±0.27** | **60.18±0.24** | **32.43±0.22** | **26.76±0.17** |

Table 11: Average experimental results of different inputs of $G_\psi$ with ResNet-18 and $\epsilon = 8/255$ in 7 runs.

| DATASET | CIFAR-10 | | | CIFAR-100 | | |
|---|---|---|---|---|---|---|
| METHOD | Natural | PGD-20 | AA | Natural | PGD-20 | AA |
| w/ **z** | 83.63±0.63 | 56.86±0.64 | 50.67±0.61 | 61.52±0.72 | 32.74±0.55 | 26.64±0.43 |
| w/ **z** + one-hot label | 83.86±0.33 | 57.27±0.39 | 50.92±0.38 | 62.05±0.32 | 32.96±0.35 | 26.87±0.28 |
| w/ **z** + logits | **84.17±0.21** | **57.55±0.15** | **51.36±0.14** | **62.57±0.17** | **33.37±0.15** | **27.11±0.15** |

### F.7 Impact of Different Inputs of $G_\psi$

For the Generative Adversarial Nets (GANs), Conditional GANs (CGANs) [41] usually have higher quality generative results than vanilla ones. Consequently, motivated by CGAN, we use a combination of the sample logits from model $f_\theta$ and stochastic sampling noise vector $\mathbf{z} \sim \mathcal{N}(\mathbf{0}, \mathbf{I})$ following a standard Gaussian distribution as input of $G_\psi$, which is different CGAN with one-hot label. To better show the effectiveness, we perform experiments on CIFAR-10 and CIFAR-100 against the PGD-20 and AA. Table 11 shows that the input of $G_\psi$ significantly influences the model's natural and robust accuracy. Moreover, the performance of input with logits has a smaller variance than in other cases, indicating the logits make the training runs more stable. As shown in Figure 9 (see page 28), the gaps between the generated amplitude spectrum of different inputs are quite small. However, the recombination from the input with $\mathbf{z}$ and logits have a small gap for benign samples, with no significantly changed sample.

### F.8 Single AE Generation of DAT

In the training procedure of DAT, we take both benign and recombined samples' AEs into AT. To reduce the time consumption of AE generation, we propose an efficient strategy to produce AEs.

We can also generate AE of recombined data by mixing the generated amplitude spectrum with that in the benign sample's AE. For an example, with $(\mathbf{x}, y) \in \mathcal{D}$, we use the $\mathbf{x}$'s AE $\mathbf{x}'$ to obtain the amplitude of recombined AE as

$$\mathcal{A}_{mix}(\mathbf{x}') = \lambda \mathcal{A}_G(\mathbf{x}') + (1 - \lambda)\mathcal{A}(\mathbf{x}'). \tag{13}$$

Then, $\hat{\mathbf{x}}'$ is obtained by

$$\hat{\mathbf{x}}' = \mathcal{F}^{-1}(\mathcal{A}_{mix}(\mathbf{x}'), \mathcal{P}(\mathbf{x}')). \tag{14}$$

Table 12: Average experimental results with single and dual AE on CIFAR-10 and CIFAR-100 with ResNet-18 and $\epsilon = 8/255$ in 7 runs.

| DATASET METHOD | CIFAR-10 | | | CIFAR-100 | | |
|---|---|---|---|---|---|---|
| | Natural | PGD-20 | AA | Natural | PGD-20 | AA |
| Single AE | 84.28±0.26 | 55.78±0.35 | 50.43±0.32 | 62.64±0.34 | 31.64±0.37 | 26.38±0.24 |
| Dual AE | **84.17±0.21** | **57.55±0.15** | **51.36±0.14** | **62.57±0.17** | **33.37±0.15** | **27.11±0.15** |

We call such a procedure to obtain recombined data and AE a single AE DAT, while the process described in the main text is Dual AE DAT. To explore the difference in performance between the two AE generation strategies, we perform experiments with the same experimental settings in Sec. E.2 on CIFAR-10 and CIFAR-100 with ResNet-18. The PGD-20 and AA are selected to evaluate the model's performance with single or dual AE generation. As shown in Table 12, we report the natural and robust results. According to these experiments, the natural accuracy gap for the DAT with single and dual AE is small, while the difference for the robust performance is large. As mentioned in the work, the amplitude spectrum of the sample influences the model's learning of the phase patterns. The single AE of DAT does not reflect the impact of mixed amplitude $\hat{\mathbf{x}}$ on adversarial perturbation, resulting in a lower robust performance than dual AE for DAT. Additionally, as shown in Table 4, although the model trained with single AE has a lower robust performance than common DAT, it still enhances the model's robustness compared to the baseline. That indicates the single AE for DAT can also enforce the model to learn more robust phase patterns.

### F.9 Limitations and Future Works

Although the proposed DAT significantly improves the model's robustness and natural accuracy of the model, there are still some limitations of the work. Since the robust over-fitting issues, DAT without AWP needs to be trained in a limited epoch, restricting its performance. Moreover, our DAT focused on protecting from adversarial attacks. Other types of image corruptions, *e.g.*, Gaussian noise, and defocus blur, can also influence the model's performance. DAT needs to be developed further to protect the model from various image corruptions. Due to the limitation of page length, we only provide the experimental results on three frequently used datasets and deep models in the main paper. In the future, we will add experiments on more large-scale datasets with different deep models.

# G Theoretical Analysis and Proof

Here we restate and prove Theorem 3.2.

**Theorem 3.2** (Weight Regularization of Amplitude Features). *Grant Assumption 3.1, when the empirical risk $\hat{R}$ is minimized with some convex loss function $\mathcal{L}$ (e.g. CE loss):*

$$\hat{R}(\mathbf{W}) \coloneqq \frac{1}{|\mathcal{D}|} \sum_{(\mathbf{x}, y) \in \mathcal{D}} \mathbb{E}_{t(\mathbf{x}) \sim \mathcal{T}(\mathbf{x})} \left[ \mathcal{L}\left( \mathbf{W}^\top \mathbf{h}(t(\mathbf{x})), y \right) \right],$$

*we have $w_{j,a} \to 0$ for all $j \in [c]$, where $w_{j,a}$ is the corresponding weights of amplitude features $h_a$.*

We consider a quadratic approximation of the objective in Theorem 3.2, and provide a further and extended analysis following [16, 26]. Using the second-order Taylor expansion, we expand each term of the objective function:

$$\mathbb{E}_{t(\mathbf{x}) \sim \mathcal{T}(\mathbf{x})} \left[ \mathcal{L}\left( \mathbf{W}^\top \mathbf{h}(t(\mathbf{x})), y \right) \right] = \mathcal{L}(\mathbf{W}^\top \bar{\mathbf{h}}, y) + \frac{1}{2} \mathbb{E}_{t(\mathbf{x}) \sim \mathcal{T}(\mathbf{x})} \left[ \Delta^\top \mathbf{H}(\zeta, y) \Delta \right], \quad (15)$$

where $\bar{\mathbf{h}} = \mathbb{E}_{t(\mathbf{x}) \sim \mathcal{T}(\mathbf{x})} \left[ \mathbf{h}(t(\mathbf{x})) \right]$, $\Delta = \mathbf{W}^\top \left( \mathbf{h}(t(\mathbf{x})) - \bar{\mathbf{h}} \right)$, and $\mathbf{H}$ is the Hessian matrix with $\zeta$ denoting the remainder. We introduce Lemma G.1 to demonstrate the properties of $\mathbf{H}$.

**Lemma G.1.** *For the CE loss with softmax, the Hessian matrix w.r.t. the loss function satisfies*

1. $\mathbf{H} \succeq 0$, *i.e.*, $\mathbf{H}$ *is semi-definite.*

2. $\mathbf{H}$ *is independent of the true label vector* $\mathbf{y}$.

*Proof.* The CE loss for a single instance $\mathbf{x}$, when given the true label vector $\mathbf{y}$ (using one-hot encoding with $c$-dimension) is defined as

$$L = -\sum_{i=1}^{c} y_i \log(p_i). \quad (16)$$

The gradient of $L$ w.r.t. the logits is computed as

$$\frac{\partial L}{\partial z_k} = p_k - y_k, \quad (17)$$

for $k \in [c]$, where $p_k = \sigma(\mathbf{z})_k$ is the predicted probability of class $k$. The Hessian matrix $\mathbf{H}$, which is the matrix of second derivatives of $L$, has elements given by

$$\frac{\partial^2 L}{\partial z_k \partial z_j} = \frac{\partial}{\partial z_j}(p_k - y_k) = \frac{\partial p_k}{\partial z_j}. \quad (18)$$

Using the derivation properties of the softmax function, the above derivative simplifies to

$$\frac{\partial p_k}{\partial z_j} = p_k(\delta_{kj} - p_j), \quad (19)$$

where $\delta_{kj}$ is the Kronecker delta, equal to 1 if $k = j$ and 0 otherwise. Thus, the Hessian matrix $\mathbf{H}$ is

$$H_{kj} = p_k(\delta_{kj} - p_j). \quad (20)$$

**Positive Semi-Definiteness.** The Hessian matrix can be expressed in matrix form as

$$\mathbf{H} = \mathbf{P}(\mathbf{I} - \mathbf{P}), \quad (21)$$

where $\mathbf{P}$ is a diagonal matrix with diagonal entries $p_k$, and $\mathbf{I}$ is the identity matrix. The product of $\mathbf{P}$ and $\mathbf{I} - \mathbf{P}$, where $\mathbf{P}$ is a diagonal matrix with entries between 0 and 1, ensures that $\mathbf{H}$ is positive semi-definite.

**Independence from Label y.** The structure and values of $\mathbf{H}$ depend solely on the predicted probabilities $\mathbf{p}$, which are functions of $\mathbf{z}$ and not dependent on the specific values of $\mathbf{y}$. This characteristic signifies that the Hessian's form is invariant w.r.t. the true label vector $\mathbf{y}$, thus highlighting its independence. $\qquad \square$

In the following, we omit the subscript of the expectation in Eq. (15) for simplicity. According to [8], when minimizing $\hat{R}$, the first term in RHS of Eq. (15) may be no room for further improvement because the local optima has nearly the same value as the global optimum in neural networks, and we put our main focus on the second term. We first derive an upper bound for the second term in Lemma G.2, since directly optimizing it which requires a third-order derivative is intractable.

**Lemma G.2** (stated informally, *cf*. Theorem 3.1 in [8]). *Assume that the covariance between $\|\mathbf{H}\|_F$ and $\|\Delta\|_2$ is zero, we can get the upper bound of the second term as*

$$\mathbb{E}\left[\Delta^\top \mathbf{H}\Delta\right] \leq \mathbb{E}\left[\|\mathbf{H}\|_F\right] \cdot \mathbb{E}\left[\|\Delta\|_2^2\right]. \tag{22}$$

*Proof.* By Hölder's inequality, we have

$$\mathbb{E}\left[\Delta^\top \mathbf{H}\Delta\right] = \mathbb{E}\left[\|\Delta\|_p \|\mathbf{H}\Delta\|_q\right] \leq \mathbb{E}\left[\|\Delta\|_p \|\mathbf{H}\|_{r,q} \|\Delta\|_r\right] = \mathbb{E}\left[\|\mathbf{H}\|_{r,q}\right] \cdot \mathbb{E}\left[\|\Delta\|_p \|\Delta\|_r\right], \tag{23}$$

where $1/p + 1/q = 1$, $\|\cdot\|_{r,q}$ is an induced matrix norm. When $p = q = r = 2$, we have

$$\mathbb{E}\left[\Delta^\top \mathbf{H}\Delta\right] \leq \mathbb{E}\left[\|\mathbf{H}\|_2\right] \cdot \mathbb{E}\left[\|\Delta\|_2^2\right] \leq \mathbb{E}\left[\|\mathbf{H}\|_F\right] \cdot \mathbb{E}\left[\|\Delta\|_2^2\right]. \tag{24}$$

$\square$

Intuitively, since $\|\mathbf{H}\|_F$ represents the sharpness/flatness of loss landscape, and $\|\Delta\|_2$ describes the translation of landscape, the two terms can be assumed independent, where their covariance can be assumed as zero [8]. Consider the latter item in RHS of Eq. (22):

$$
\begin{aligned}
\mathbb{E}\left[\|\Delta\|_2^2\right] &= \mathbb{E}\left[\Delta^\top \Delta\right] \\
&= \mathbb{E}\left[\left(\mathbf{h}(t(\mathbf{x})) - \bar{\mathbf{h}}\right)^\top \mathbf{W}\mathbf{W}^\top \left(\mathbf{h}(t(\mathbf{x})) - \bar{\mathbf{h}}\right)\right] \\
&= \mathbb{E}\left[\mathrm{tr}\left(\mathbf{W}\mathbf{W}^\top \left(\mathbf{h}(t(\mathbf{x})) - \bar{\mathbf{h}}\right)\left(\mathbf{h}(t(\mathbf{x})) - \bar{\mathbf{h}}\right)^\top\right)\right] \\
&= \mathrm{tr}\left(\mathbf{W}\mathbf{W}^\top \mathbb{E}\left[\left(\mathbf{h}(t(\mathbf{x})) - \bar{\mathbf{h}}\right)\left(\mathbf{h}(t(\mathbf{x})) - \bar{\mathbf{h}}\right)^\top\right]\right) \\
&= \mathrm{tr}\left(\mathbf{W}\mathbf{W}^\top \boldsymbol{\Sigma}\right),
\end{aligned} \tag{25}
$$

where $\boldsymbol{\Sigma}$ is the covariance matrix of $\mathbf{h}(t(\mathbf{x}))$. For a model trained well enough, the feature extractor can completely distinguish different features, *i.e.*, $\boldsymbol{\Sigma}$ is a diagonal matrix. Then we can obtain

$$\mathrm{tr}\left(\mathbf{W}\mathbf{W}^\top \boldsymbol{\Sigma}\right) = \mathrm{tr}\left(\sum_{i=1}^c \mathbf{w}_i \mathbf{w}_i^\top \boldsymbol{\Sigma}\right) = \sum_{i=1}^m \sum_{j=1}^c w_{j,i}^2 \mathrm{Var}\left[h_i(t(\mathbf{x}))\right], \tag{26}$$

where $w_{j,i}$ is the $j$-th entry of $\mathbf{w}_i$. If the variance of the feature $h_i(t(\mathbf{x}))$ is large, minimizing the empirical risk $\hat{R}$ requires $\sum_{j=1}^c w_{j,i}^2 \to 0 \implies w_{j,i} \to 0$ for all $j \in [c]$. By Assumption 3.1, it is trivial that $\mathrm{Var}[h_a(t(\mathbf{x}))] > \mathrm{Var}[h_p(t(\mathbf{x}))]$ and therefore $w_{j,a} \to 0$. Under this circumstance, the model is enforced to learn more phase information. The claim follows.

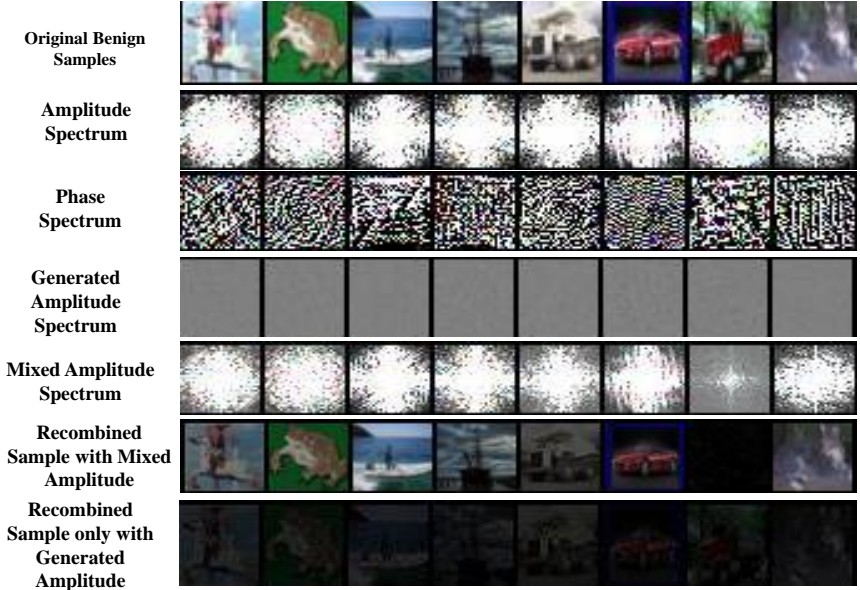

Figure 8: Visualization of recombined samples w/wo mix-up.

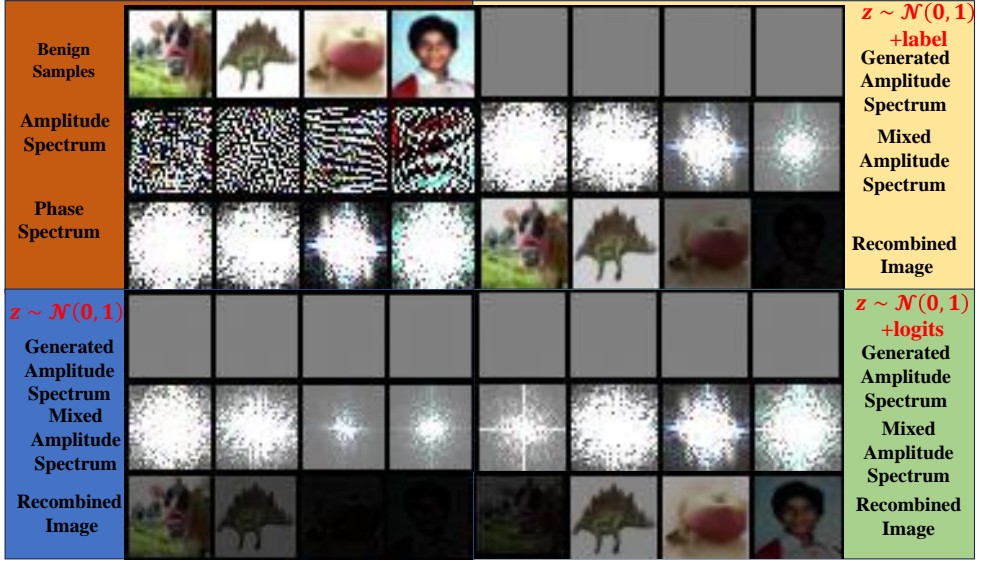

Figure 9: Visualization of recombined samples based on the amplitude spectrum generated by different inputs of AAG.

