# OpenReview forum: "DAT: Improving Adversarial Robustness via Generative Amplitude Mix-up in Frequency Domain"
_NeurIPS.cc/2024/Conference — NeurIPS 2024 poster_

### Official Review · Reviewer_FTsi · 2024-06-25

**Soundness:** 3
**Presentation:** 2
**Contribution:** 3
**Rating:** 6
**Confidence:** 3

**Summary:**

Motivated by observation on the difference influence of amplitude and phase of adversarial examples, this paper propose a framework to generate better adversarial examples for adversarial training. Experiments verify the effectiveness of the proposed approach.

**Strengths:**

1. The motivation in Figure is clear and solid.
2. The experimental improvements are clear and consistent.

**Weaknesses:**

1. The proposed framework contain many modules, making it complex and difficult to implement.
2. Some descriptions are unclear and some details are missed.

See questions for details.

**Questions:**

1. In Figure 1, the authors claim that “The adversarial perturbation severely damages phase pattern and the frequency spectrum, while amplitude patterns are rarely impacted.” However, this comparison is not clear. It seems the amplitude patterns still changes a lot in Figure 1. Both the patterns and spectrum change. Could you provide more proof on this observation (e.g., results on more images and more datasets)?
2. In Line 88, the features induced from the amplitude and phase patterns of x are denoted as $h_a(\mathbf{x})$ and $h_p(\mathbf{x})$ respectively.  Does it indicate an assumption that some parts of the learned $h$ correspond to the amplitude features? Do we need carefully design the network to explicitly accomplish this? It also makes the theoretical analysis in Sec. 3.4 difficult to understand.
3. As the proposed framework contains multiple modules and optimization steps, it is surprising that the additional time consumptions are small. Could you please kindly provide more explanation on this? Do we need any relatively complex optimization tricks? Furthermore, what is the memory cost during training and could you please present the corresponding comparison with the baseline methods?
4. It seems the mix-up operation is important in the whole framework. The adversarial amplitude and original amplitude are mixed by linear combination and the proportion is sampled from a uniform distribution. How do you make such a design? Have you ever tried other possible methods?

**Limitations:**

The authors discuss the limitations of this work in Sec. F.9 and say that the verification may be insufficient. No limitation of the proposed method itself is discussed.

---

> ### Author Rebuttal · Authors · 2024-08-06
>
> Thank you for your thoughtful review. We have provided our detailed responses below.
>
> ---
>
> **W1: Complexity**
>
> **R1:** Our DAT comprises **only 2** modules: the trained model and the adversarial amplitude generator (AAG). Due to the AAG's simple four-layer linear architecture, DAT's time consumption remains comparable to existing methods. Foremost, as shown in **Table C.1**,  DAT significantly enhances model robustness against various adversarial attacks.
>
> **Table C.1: Time Consumption and Memory Cost on CIFAR-10 with ResNet-18**
> ||PGD-AT|TRADES|ST|SCARL|DAT (Ours)
> -|:-:|:-:|:-:|:-:|:-:
> AE Generation Time (s)|155|155|275|166|157
> Optimization Time (s)|32|32|45|55|61
> Memory (MB)|2589|3586|4589|5836|5763
>
> ---
>
> **W2: Unclear Descriptions and Missing Details**
>
> **R2:** Please see the answers below.
>
> ---
>
> **Q1: Impact of Adversarial Perturbation on Amplitude and Phase**
>
> **A1:** In Figure 1, the phase (3rd column) and amplitude (5th column) spectra are affected by adversarial perturbation. Moreover, the phase patterns (2nd column), particularly within the red rectangular, are damaged, while the amplitude patterns (4th column) are rarely affected. Since models rely on patterns, especially semantics in phase patterns [i], to make predictions, AE with damaged phase patterns hinder accurate prediction. Further results can be found in **Figures 1-4** of the **attached PDF** within the global **Author Rebuttal**.
>
> [i] Yin et al. A Fourier perspective on model robustness in computer vision. NeurIPS 2019.
>
> ---
>
> **Q2: Learned Features and Theoretical Analysis**
>
> **A2:**
>
> * **Learned Features:** Indeed, we assume $h$ can extract features amplitude and phase patterns, respectively, since it is a **widely accepted assumption within feature learning theory [j, k, l]**.  In practice, $h$ can extract phase patterns (semantics) and amplitude patterns (color and style) respectively [i, l],  which can be readily **achieved using general networks in conjunction with the discrete Fourier transform, eliminating the need for specifically defined networks.** Such decoupling methods are commonly adopted. For instance, [i] represents Fourier information as amplitude and phase patterns, while [m] has demonstrated that CNNs (e.g., ResNet-50) can inherently capture these amplitude and phase patterns without meticulously designed components.
>
>  * **Sketch of Theoretical Analysis:** Given that DAT utilizes AAG to generate adversarial amplitude, which is then combined with the benign image's phase to augment data, it naturally follows that during AT, amplitude differences between augmented and original datasets are more pronounced than phase differences (Assumption 3.1).  As a result, to achieve convergence during empirical risk minimization, the model regularizes the weights corresponding to amplitude features (Theorem 3.2), thereby reducing their influence on predicted labels (Corollary 3.3), and consequently shifting the focus towards phase features. We hope this explanation addresses your questions.
>
> [j] Ilyas et al. Adversarial examples are not bugs, they are features. NeurIPS 2019.
>
> [k] He et al. Data augmentation revisited: Rethinking the distribution gap between clean and augmented data. arXiv 2019.
>
> [l] Xu et al. A Fourier-based framework for domain generalization. CVPR 2021.
>
> [m] Chen et al. Amplitude-phase recombination: Rethinking robustness of convolutional neural networks in frequency domain. ICCV 2021.
>
> ---
>
> **Q3: More Explanation on Time and Memory Costs**
>
> **A3:** For clearer comparison, we separately analyze the time consumption for AE generation and model optimization (updating the model's parameters), and present the memory costs in **Table C.1**. As shown, the majority of time consumption in DAT is attributed to AE generation. Thanks to our efficient AE generation strategy, DAT exhibits comparable time consumption to baselines. Additionally, the AAG, a four-layer linear model, demands little training time. Since both benign and recombined samples, along with their AEs, are used in model optimization, DAT takes roughly twice the time for optimization compared to baselines,  e.g., PGD-AT and TRADES.
>
> ---
>
> **Q4: Mix-up Operation Design**
>
> **A4:**
>
> * **Linear Mix-up:** The linear mix-up operation is designed to preserve the energy of the amplitude spectrum and maintain the original amplitude information, since linear mix-up is a simple and effective method for augmenting data with less impact on the sample's original information [n, o]. Otherwise, the original amplitude could be compromised, thereby hindering accurate model predictions.
>
> * **$\lambda$ Distribution:** To investigate the influence of $\lambda$, we conduct experiments on CIFAR-10 using ResNet-18 with $\lambda$ sampled from $\mathrm{Uniform}(0,1)$, $\mathrm{Beta}(1,1)$, and $\mathcal{N}(0,1)$. Given that $\mathcal{N}(0,1)$ can yield negative values, we transform $\lambda$ into $\mathrm{sigmoid}(\lambda)$. **Table C.2** suggests that the distribution type of $\lambda$ has little impact on the model's robust and natural performance.
>
> **Table C.2: DAT with Different $\lambda$ Distributions**
> Distribution|Natural|AA
> -|:-:|:-:
> Uniform|84.17%|51.36%
> Beta|84.06%|51.24%
> Gaussian|84.09%|51.28%
>
> [n] Yun et al. Cutmix: Regularization strategy to train strong classifiers with localizable features. CVPR 2019.
>
> [o] Berthelot et al. Mixmatch: A holistic approach to semi-supervised learning. NeurIPS 2019.
>
> ---
>
> **Discussion of Limitations**
>
> **R3:** Due to robust overfitting, DAT without AWP needs to be trained for a limited number of epochs, which restricts its performance. Moreover, while DAT primarily focuses on defending against adversarial attacks, other forms of image corruption (e.g., Gaussian noise and defocus blur) can also affect the model's performance. DAT requires further development to strengthen its protection against a broader spectrum of image corruptions. A comprehensive discussion will be included in the final version of the paper.
>
> ---

---

> ### Comment · Reviewer_FTsi · 2024-08-10
>
> The rebuttal mostly addresses my concerns. Therefore, I decide to increase the score.

---

> ### Author Response · Authors · 2024-08-10
>
> Dear Reviewer，
>
> We greatly appreciate your responses and score improvements. Also, the final version of the paper will be revised based on your comments.

---

### Official Review · Reviewer_KucH · 2024-07-13

**Soundness:** 3
**Presentation:** 3
**Contribution:** 3
**Rating:** 5
**Confidence:** 5

**Summary:**

The paper introduces Dual Adversarial Training (DAT). This method enhances deep neural network resilience against adversarial attacks by employing generative amplitude mix-up in the frequency domain, focusing the model on phase patterns less impacted by such perturbations, and presenting an optimized adversarial amplitude generator for balancing robustness improvements with phase pattern retention. Experiments validate DAT's effectiveness against diverse adversarial attacks.

**Strengths:**

1. The paper is well-organized and easy to follow.
2. The experimental results show the effectiveness of the proposed method.

**Weaknesses:**

1. In the experiment, only decision space attacks are used to test the defense capability. Compared with decision space attack, feature space attack can better destroy the semantic information of benign images and obtain more powerful adversarial samples. It is recommended that some feature space attacks be added to further demonstrate the effectiveness of the proposed method, like [a] and [b].

[a] StyLess: Boosting the Transferability of Adversarial Examples, CVPR 2023

[b] Enhancing the Self-Universality for Transferable Targeted Attacks, CVPR 2023

2. The types of models used in the experiment are not rich enough. Line 258 of this paper states that "ResNet-18, WideResNet-34-10 (WRN-34-10), and WideResNet-28-10 (WRN-28-10) are used as the backbones". Line 269 says "Table 1 displays the results on CIFAR-10, CIFAR-100, and tiny-ImageNet using ResNet-18". These models are all based on ResNet architecture and have high structural similarities. To show that the method proposed in this paper is generalizable, the proposed method should be able to achieve good results on different surrogate and target models. Therefore, it is recommended to add more surrogate and target models, such as Inception_v3, ViT, etc.

3. This paper uses CIFAR-10, CIFAR-100, and Tiny ImageNet for experiments, but does not use ImageNet. However, ImageNet is the most widely used dataset in the CV field. The authors should provide additional experimental results on ImageNet for this paper's method.

**Questions:**

1. Can the authors provide some results against feature space attacks?

2. Can the authors provide some results on other models and datasets, like ImageNet and ViTs?

**Limitations:**

The authors have discussed the limitations of the method proposed in this paper. It is suggested that the authors provide some results on other models and datasets, like ImageNet and ViTs.

---

> ### Author Rebuttal · Authors · 2024-08-06
>
> Thank you for your thorough and detailed reviews. Please find our responses below.
>
> ---
>
> **W1: Additional Experiments against Feature Space Attacks**
>
> **R1:** In **Table B.1** (more results in **Tables 1-4** of **attached PDF**), we present results assessing the defense capability against feature space attacks [a, b].  We select StyLess+MTDI (SMTDI) and StyLess+MTDSI (SMTDSI) from [a], along with DTMI-CE-SU (DCS) and DTMI-Logit-SU (DLS) target to class 1 from [b], and perform experiments on ImageNette as in [c]. We adopt surrogate ResNet-50 $\rightarrow$ targets [DenseNet-121, ViT-S] and surrogate DenseNet-121 $\rightarrow$ targets [ResNet-50, ViT-S] execute black-box transfer attacks. AT on DenseNet-121 and ResNet-50 follow the same settings as Tiny-ImageNet in the paper, while ViT-S is trained adversarially as in [c]. **For the challenging SMTDSI attack, compared to the previous SOTA ST, our proposed DAT an average robustness improvement of approximately 4.9% across all four scenarios**. Moreover, since DAT performs AT in pixel space, its performance against feature space attacks is currently less satisfactory[d], which we plan to improve in the future.
>
>  **Table B.1: Model's Performance against $\ell_{\infty}$ Threat with $\epsilon=\frac{16}{255}$  of Feature Space Attacks**
> Method|SMTDI|SMTDSI|DCS|DLS
> -|:-:|:-:|:-:|:-:
> **ResNet-50$\rightarrow$DenseNet-121**
> ST|36.46%|31.68%|76.61%|73.67%
> SCARL|36.15%|31.31%|76.23%|73.46%
> DAT (Ours)|**41.62**%|**37.61**%|**79.12**%|**76.41**%
> **ResNet-50$\rightarrow$ViT-S**
> ST|46.42%|42.34%|86.27%|76.53%
> SCARL|46.32%|42.31%|86.71%|77.26%
> DAT (Ours)|**51.83**%|**46.76**%|**88.45**%|**80.41**%
> **DenseNet-121$\rightarrow$ResNet-50**
> ST|38.67%|33.29%|78.93%|66.83%
> SCARL|38.17%|33.13%|78.70%|65.93%
> DAT (Ours)|**43.38**%|**38.27**%|**82.12**%|**67.49**%
> **DenseNet-121$\rightarrow$ViT-S**
> ST|47.53%|43.07%|87.83%|78.41%
> SCARL|47.24%|42.95%|87.40%|77.58%
> DAT (Ours)|**52.06**%|**47.43**%|**89.12**%|**79.41**%
>
> [a] Liang and Xiao. StyLess: boosting the transferability of adversarial examples. CVPR 2023.
>
> [b] Wei et al. Enhancing the self-universality for transferable targeted attacks. CVPR 2023.
>
> [c] Mo et al. When adversarial training meets vision transformers: Recipes from training to architecture. NeurIPS 2022.
>
> [d] Xu et al. Towards feature space adversarial attack by style perturbation. AAAI 2021.
>
> ---
>
> **W2: Additional Experiments with More Backbones**
>
> **R2:** In **Table B.2**, we provide results on CIFAR-10 with Inception\_v3, DenseNet-121, and ViT-S. The settings for Inception\_v3, and DenseNet-121 remain consistent with those mentioned in the paper, while ViT-S follows [c]. Compared to the previous SOTA ST, **DAT achieves robustness improvements of 0.63%, 0.90%, and 0.61%** against AA across various backbones, demonstrating its versatility and effectiveness on a range of deep models.
>
>  **Table B.2: Experimental Results with More Backbones against $\ell_{\infty}$ Threat with $\epsilon=\frac{8}{255}$**
> ||PGD-AT||TRADES||MART||ST||SCARL||DAT (Ours)||
> -|:-:|:-:|:-:|:-:|:-:|:-:|:-:|:-:|:-:|:-:|:-:|:-:
> Backbone|Natural|AA|Natural|AA|Natural|AA|Natural|AA|Natural|AA|Natural|AA
> Inception\_v3|85.26%|48.83%|86.38%|49.74%|83.41%|48.75%|86.75%|51.18%|84.43%|50.98%|**87.74**%|**51.81**%
> DenseNet-121|86.34%|51.24%|86.92%|52.03%|84.11%|50.92%|58.63%|53.26%|85.15%|53.12%|**88.51**%|**54.16**%
> ViT-S|81.86%|47.33%|81.95%|48.45%|80.31%|47.13%|82.42%|49.71%|80.85%|49.67%|**83.15**%|**50.32**%
>
> ---
>
> **W3: Additional Experiments on ImageNet**
>
> **R3:** In **Table B.3**, we show results on ImageNet-1K, which adopt settings and baselines' results from [h]. It is worth noting that existing AT methods with multiple iteration steps for AE generation do not provide results on ImageNet-1K, since AT with a 10-step AE generation on ImageNet-1K needs **approximately one week**. Due to single iterative step AE generation using less time, some fast AT methods based on FGSM [e, f, g, h] typically perform experiments on ImageNet-1K. Consequently, considering the time limitation of rebuttal, to satisfy your suggestion of providing results on ImageNet-1K, we combine DAT with several fast AT methods [e, f, g, h] and compare its performance with these methods. **For the previous SOTA FGSM-PGK, DAT combined with FGSM-PGI obtains robustness improvements of 2.1% and 1.4% against PGD-10 and PGD-50**, respectively.
>
> **Table B.3: Experimental Results on ImageNet-1K with ResNet-50 against $\ell_{\infty}$ Threat with $\epsilon=\frac{4}{255}$**
> Method|Natural|PGD-10|PGD-50
> -|:-:|:-:|:-:
> PGD-AT|59.19%|35.87%|35.41%
> Free-AT (m=4) [e]|63.42%|33.22%|33.08%
> FGSM-RS [f]|63.65%|35.01%|32.66%
> FGSM-PGI [g]|64.32%|36.24%|34.93%
> FGSM-PGK [h]|66.24%|37.13%|35.70%
> DAT+Free-AT (m=4) (Ours)|**66.36**%|**36.27**%|**36.12**%
> DAT+FGSM-RS (Ours)|**66.49**%|**37.43**%|**35.76**%
> DAT+FGSM-PGI (Ours)|**67.12**%|**39.24**%|**37.13**%
>
> [e] Shafahi et al. Adversarial training for free!. NeurIPS 2019.
>
> [f] Wong et al. Fast is better than free: Revisiting adversarial training. ICLR 2020.
>
> [g] Jia et al. Prior-guided adversarial initialization for fast adversarial training. ECCV 2022.
>
> [h] Jia et al. Improving fast adversarial training with prior-guided knowledge. TPAMI 2024.
>
> ---
>
> **Q1: Results against Feature Space Attacks**
>
> **A1:** Please refer to **Table B.1** in **R1** and **Tables 1-4** in the **attached PDF**.
>
> **Q2: Results on Other Models and Datasets**
>
> **A2:** For results on additional backbones, please see **Table B.2** in **R2**. For experiments on ImageNet-1K, kindly refer to **Table B.3** in **R3**.
>
> ---

---

> > ### Author Response · Authors · 2024-08-14
> > **Follow-up on Submission1111 Rebuttal - Urgent Request for Feedback**
> >
> > Dear Reviewer,
> >
> > We are truly sorry to trouble you again, but with **less than 8 hours** left before the rebuttal discussion period ends at ***11:59 PM AoE today***, we are writing with a humble and heartfelt request for your feedback on our rebuttal. We have put our utmost effort into addressing your invaluable comments and suggestions. We would be deeply grateful if you could take a few moments to review our responses and share any further thoughts. If our responses are satisfactory, we would be sincerely thankful if you might consider reflecting this in your review score.
> >
> > We greatly value your guidance, and we are genuinely appreciative of your time and consideration.
> >
> > Thank you so much for your continued support.
> >
> > Best regards,
> > Submission1111 Authors

---

> > ### Comment · Reviewer_KucH · 2024-08-14
> >
> > Thank you for your responses!
> >
> > For W1. could you please explain why you use a different dataset and a different  threat model (a black-box setting) than those of Table 1 in the main paper?
> >
> > For W3, could you please explain why you use different attacks than those of Table 1 (FGSM, PGD-20, PGD-100, C\&W, and AA)  in the main paper?

---

> ### Author Response · Authors · 2024-08-13
> **Follow-up on Submission1111 Rebuttal Response**
>
> Dear Reviewer,
>
> We hope you are doing well. We sincerely appreciate the detailed feedback and the time you have devoted to reviewing our submission.
>
> In our recent rebuttal, we carefully address your concerns and provide detailed responses:
>
> - **Additional Experiments on Feature Space Attacks**: We conduct experiments on several **feature space attacks**, specifically including methods like **SMTDI/SMTDSI [a] and DCS/DLS [b]** as you suggested, to demonstrate the robustness of our approach against more challenging adversarial attacks. These results show that our method performs better than previous SOTA techniques in most scenarios, despite being originally designed for pixel space attacks.
>
> - **Incorporating Diverse Backbones**: We expand our experiments to include **additional backbones such as Inception_v3 and ViT-S**, demonstrating that our method generalizes well across different network architectures. The results confirm that our approach maintains its effectiveness and robustness across these diverse models.
>
> - **Extensive Experiments on ImageNet-1K**: We also provide **extensive results on the ImageNet-1K dataset**, a critical benchmark in the CV field. Given the time constraints, we combine our method with fast adversarial training techniques like FGSM-PGK and FGSM-PGI, achieving significant improvements in robustness compared to the baseline methods.
>
> These comprehensive experiments are conducted with great care to address your concerns about the scope and applicability of our approach, and to demonstrate its generalization and robustness across different models and datasets.
>
> With the rebuttal discussion period closing tomorrow, ***Aug 13, 11:59 PM AoE***, we would deeply appreciate your input if there are any further questions or points that you feel need clarification. If our rebuttal has satisfactorily addressed your concerns, we humbly request that you consider reflecting this positively in your rating score.
>
> Thank you very much for your continued support and thoughtful consideration.
>
> Best regards,
> Submission1111 Authors

---

> > ### Author Response · Authors · 2024-08-13
> > **Follow-up on Submission1111 Rebuttal - Request for Timely Feedback**
> >
> > Dear Reviewer,
> >
> > We hope this message finds you well. We apologize for reaching out again, but with the rebuttal discussion period **ending in less than 24 hours** on ***Aug 13, 11:59 PM AoE***, we kindly request your timely feedback on our rebuttal.
> >
> > To briefly restate our previous points in our rebuttal:
> >
> > - We conduct additional experiments on **feature space attacks**, including the SMTDI/SMTDSI [a] and DCS/DLS [b] methods that you suggested. These experiments demonstrate the robustness of our approach against more challenging adversarial attacks.
> >
> > - We expand our experiments to incorporate **diverse backbones**, such as Inception_v3 and ViT-S, showing that our method generalizes well across different network architectures.
> >
> > - We provide **extensive results on the ImageNet-1K dataset**, which is a critical benchmark in the CV field. Our results show significant improvements in robustness compared to baseline methods.
> >
> > These comprehensive efforts are made with great care to thoroughly address your concerns and demonstrate the generalization and robustness of our approach across different models and datasets.
> >
> > If our rebuttal has satisfactorily addressed your concerns, we humbly request that you consider reflecting this positively in your rating score. We would be deeply grateful for your timely input.
> >
> > Thank you very much for your continued support and thoughtful consideration.
> >
> > Best regards,
> > Submission1111 Authors

---

> ### Author Response · Authors · 2024-08-14
> **Re: Official Comment by Reviewer KucH**
>
> Dear Reviewer,
>
> We are truly grateful for your continued time and effort in further discussing our submission. Below is our detailed response to the two questions you raised.
>
> ---
>
> > ***"For W1. could you please explain why you use a different dataset and a different threat model (a black-box setting) than those of Table 1 in the main paper?"***
>
> + **Dataset:**
> The training dataset used in [a] and [b] is ImageNet-1k. We understand your concern and would like to clarify that existing adversarial training methods with multiple iteration steps for AE generation do not report results on ImageNet-1K, since AT with a 10-step AE generation on ImageNet-1K requires **approximately one week**. Additionally, these feature space attacks need to be performed on real-world dataset, e.g. ImageNet. Given these considerations, we choose ImageNette, a widely used subset of ImageNet-1K in adversarial training [c], for the experiments in **W1**.
>
> + **Backbones:**
> In [a] and [b], multiple backbones are adopted, including ResNet-50 and DenseNet-121. To better demonstrate the robust generalization of our method against feature space attacks, we select ViT-S, which **you mentioned in W2**, for experiments in **W1**. It is important to note that baselines in **W1** do not provide experimental results and settings with so many backbones, meaning we need significant time to explore these methods' settings across different backbones. Furthermore, given the rebuttal's **6000 characters** limit, it is challenging to present results for so many backbones. Consequently, to balance these factors and provide a clear comparison, combining the considerations of both **W1** and **W2**, we select **ResNet-50, DenseNet-121, and ViT-S** for **W1**.
>
> ---
>
> > ***"For W3, could you please explain why you use different attacks than those of Table 1 (FGSM, PGD-20, PGD-100, C&W, and AA) in the main paper?"***
>
> The adversarial attacks we evaluate in **W3** follow [h]. To ensure a fair and consistent comparison on ImageNet-1K, we use these specific adversarial attacks to provide the results in  **W3**.
>
> ---
>
> We sincerely hope these clarifications address your concerns. If there are any further questions or if you require additional explanations, we are more than happy to provide them. We also hope that if our responses sufficiently address your concerns, you might consider reflecting this positively in your review score.

---

### Official Review · Reviewer_8ttu · 2024-07-13

**Soundness:** 3
**Presentation:** 3
**Contribution:** 3
**Rating:** 6
**Confidence:** 4

**Summary:**

This paper investigates a novel approach to improving adversarial training by performing data augmentation in the frequency domain. The authors propose a unique pipeline that jointly optimizes a classification network and a generator network. The generator is used to create adversarial noise, which is added to the amplitude component of the input to encourage the model to more accurately capture phase patterns. The authors conduct a comprehensive evaluation of their approach across multiple datasets and architectures, demonstrating its competitive performance.

**Strengths:**

Comprehensive experiments on several datasets with different architectures.

A novel approach that improves adversarial training by performing data augmentation using a generator network in the frequency domain.

**Weaknesses:**

In Table 1 and Table 2, the authors have only compared their approach with a few weak baselines. Since the proposed approach is based on data augmentation, the authors should consider comparing it with other data augmentation-based approaches, such as DAJAT[1] and [47]. Although DAJAT’s results are presented in Table 3, the table does not show the clean accuracy of the models, making it difficult to assess the accuracy-robustness tradeoff. In fact, DAJAT’s clean accuracy is significantly higher than that of the proposed approach. For instance, DAJAT’s clean accuracies on CIFAR-10 and CIFAR-100 (with ResNet-18) are 86.67% and 66.96%, respectively, while the proposed approach’s are 84.17% and 63.28%. [37]’s clean accuracy and AA robust accuracy with WRN-34-10 are 86.18% and 58.09%, respectively, while the proposed approach’s with WRN-34-10 are 86.78% and 56.46%.

The proposed pipeline is complicated, with a multi-part loss function and a requirement for joint optimization with a generator network. This complexity may necessitate extensive hyperparameter tuning to achieve satisfactory performance, making it a much more complex error-prone solution compared to data augmentation approaches such as [1] and [47].

**Questions:**

No questions

**Limitations:**

Yes

---

> ### Author Rebuttal · Authors · 2024-08-06
>
> Thank you for your review. Please find our responses below.
>
> ---
>
> **W1: Natural Accuracy Compared to DAJAT [1]**
>
> **R1:** Our DAT utilizes **only 1** augmentation per benign training sample to prioritize speed and simplicity in this paper, while **DAJAT applies **2** and **3** data augmentations integrating AWP, SWA, and variable $\epsilon$ and $\alpha$** to achieve the natural performance you mentioned. Moreover, DAT exhibits superior robustness relative to DAJAT, albeit with a trade-off in natural accuracy. To ensure a fair comparison under identical training settings, we employ the adversarial amplitude generator in DAT to produce 3 adversarial amplitudes, resulting in 3 recombined data for each benign sample. Using these 3 recombined data, we extend our DAT to include 2 and 3 augmentations, integrating variable $\epsilon$ and $\alpha$, AWP, and SWA as in DAJAT [1]. Furthermore, to illustrate the trade-off between natural and robust accuracy, we select the checkpoint with the best performance on natural validation data. As illustrated in **Table A.1**, where DAJAT's performance is selected from [1], with 110 and 200 training epochs, we present a comparison of natural accuracy and robustness (AA) between the two methods on CIFAR-10 and CIFAR-100 with ResNet-18. When evaluated under the same augmentation conditions and experimental settings, our DAT outperforms DAJAT in both robustness and natural accuracy, with a smaller increase in time consumption per training epoch.
>
> **Table A.1: Comparison of DAJAT and DAT with the Same Settings**
> |||DAJAT|||DAT (Ours)|||
> -|:-:|:-:|:-:|:-:|:-:|:-:|:-:
> Dataset|#Aug.|Natural|AA|Time (s/epoch)|Natural|AA|Time (s/epoch)
> **110 training epochs**
> CIFAR-10|2|85.99%|51.48%|295|**86.38**%|**51.92**%|318
> CIFAR-10|3|86.67%|51.56%|383|**86.81**%|**52.13**%|407
> CIFAR-100|2|66.84%|27.32%|300|**67.12**%|**27.95**%|323
> CIFAR-100|3|66.96%|27.62%|407|**67.43**%|**28.16**%|429
> **200 training epochs**
> CIFAR-10|2|85.71%|52.50%|295|**86.18**%|**53.16**%|318
> CIFAR-10|3|86.24%|52.66%|383|**86.63**%|**53.31**%|407
> CIFAR-100|2|65.45%|27.69%|300|**66.32**%|**28.35**%|323
> CIFAR-100|3|65.63%|27.92%|407|**66.74**%|**28.66**%|429
>
> ---
>
> **W2: Robust Accuracy Compared to [47]**
>
> **R2:** We infer that you are referring to Rebuffi et al. [47] instead of ST [37]. In our DAT, to achieve a better trade-off between the model's performance and training time consumption, for results of DAT in Table 2, we use **only 5** iteration steps for AE generation of both benign and recombined samples, whereas [47] employs **10** iteration steps for AE generation of both benign and augmented samples with SWA. As reflected in **Table A.2**, when integrated with AWP and SWA, DAT delivers superior natural and robust performance compared to [47] with a 5-step iteration, which uses about half of training time consumption than [47] as shown in **Table A.3**. Moreover, to facilitate a fair comparison with [47], we conduct experiments on CIFAR-10 using a 10-step iteration for DAT's AE generation.  With identical iteration steps, **our DAT achieves an approximate 0.32\% improvement in robustness over [47]. For DAT with AWP and SAW, extending the iteration step to 10 can enable it to secure approximately 1\% improvement in robustness and 2.1\% in natural accuracy over [47]**, thus highlighting a more favorable trade-off between generalization and robustness.
>
> **Table A.2: Comparison between [47] and DAT with WRN-34-10**
> Method|#Iter. Step|Natural|AA
> -|:-:|:-:|:-:
> [47]|10+10|86.18%|58.09%
> DAT (Ours)|5+5|**86.78**%|**56.46**%
> DAT (Ours)|10+10|**86.23**%|**58.41**%
> DAT+AWP+SWA (Ours)|5+5|**88.65**%|**58.12**%
> DAT+AWP+SWA (Ours)|10+10|**88.28**%|**59.07**%
>
> **Table A.3: Comparison of Time Consumption per Epoch with WRN-34-10 on CIFAR-10**
> Method|#Iter. Step|Time (s)
> -|:-:|:-:
> [47]|10+10|2884
> DAJAT [1]|5+5|1532
> DAT (Ours)|5+5|1551
>
> ---
>
> **W3: Complexity**
>
> **R3:**
> * **For DAJAT [1]:**
> Our proposed DAT involves **only 2** hyperparameters, $\beta$, and $\omega$, which need to be explored by experiments.
> In contrast, **DAJAT involves additional complexity beyond the balance parameters $\beta$ and $\omega$; it requires careful scheduling for adjusting adversarial perturbation $\epsilon$, step size $\alpha$, and iteration steps, due to its use of variable $\epsilon$, $\alpha$**, and iteration steps in AT. Moreover, the schedules for tuning adversarial perturbation $\epsilon$, step size $\alpha$, and iteration steps need to be adjusted continuously throughout the entire training epoch in DAJAT.
>
> * **For [47]:**
> In the case of [47], a combination of data augmentation strategies is employed alongside the traditional TRADES method with **10** steps for AE generation and variable $\alpha$, whereas our proposed DAT requires **only 5** steps per sample to generate AE. Consequently, as shown in **Table A.3**, **[47] demands nearly double the training time compared with our DAT when augmentations are applied to benign samples**.
>
> Despite its multi-component structure and joint optimization, our method remains simpler than DAJAT and [47] in terms of both hyperparameter tuning and computational complexity.
>
> ---

---

> > ### Author Response · Authors · 2024-08-13
> > **Follow-up on Submission1111 Rebuttal Response**
> >
> > Dear Reviewer,
> >
> > I hope this message finds you well. We sincerely appreciate the time and effort you invest in reviewing our submission.
> >
> > In our recent rebuttal, we carefully address your feedback and provide detailed responses to your concerns:
> >
> > - **Expanded Comparison with Data Augmentation-Based Approaches**: We conduct additional experiments to ensure a **fair comparison** with more data augmentation-based methods, specifically including **DAJAT and Rebuffi et al. [47]**. We provide an in-depth analysis demonstrating that our approach not only **enhances robustness** but also achieves **comparable or better natural accuracy** under similar conditions.
> >
> > - **Comprehensive Analysis of Accuracy-Robustness Trade-off**: We offer a detailed examination of both **natural and robust accuracy**, showing how our method **balances these two metrics effectively**. For instance, our method **outperforms DAJAT in robustness** while maintaining **competitive natural accuracy**, as outlined in our comparison tables.
> >
> > - **Clarification on Pipeline Complexity**: We clarify concerns regarding the **complexity of our pipeline**, explaining that while our method involves multiple components, it is designed to require **fewer hyperparameters and simpler optimization steps** than comparable methods. We also demonstrate that our approach is **more efficient** in terms of **training time and computational cost**, particularly when compared to methods like **[47]**, which require longer iteration steps for adversarial example generation.
> >
> > As the rebuttal discussion period ends tomorrow, ***Aug 13, 11:59 PM AoE***, we would be truly grateful if you could let us know if there are any remaining questions or concerns regarding our submission. If our responses have satisfactorily addressed your concerns, we kindly hope you might consider reflecting this in your review score by raising it.
> >
> > Thank you once again for your thoughtful feedback and for considering our request.
> >
> > Best regards,
> > Submission1111 Authors

---

> > ### Comment · Reviewer_8ttu · 2024-08-13
> > **Type of augmentations**
> >
> > Thank the authors for the additional experiments. Could the authors clarify what do they mean by "DAJAT applies 2 and 3 data augmentations"? Additionally, in the statement "we extend our DAT to include 2 and 3 augmentations", what are the specific types of augmentations used?

---

> ### Author Response · Authors · 2024-08-13
> **Re: Type of augmentations**
>
> Dear Reviewer,
>
> Thank you very much for your prompt response and for your continued engagement with our submission. We appreciate the opportunity to clarify our approach, and below we provide detailed responses to the two points you raised.
>
> ---
>
> > ***"DAJAT applies 2 and 3 data augmentations"***
>
> Regarding your question on DAJAT [1], the natural accuracy that you referenced is achieved by generating 2 or 3 augmented data samples for each benign sample using the **AutoAugment** strategy. These benign and augmented samples, along with their corresponding adversarial examples, are then utilized in the training process.
>
> ---
>
> > ***"we extend our DAT to include 2 and 3 augmentations"***
>
> In our Dual Adversarial Training (DAT) approach, the recombined data with mixed amplitude spectra serve as augmented samples for each benign sample. **To ensure a fair comparison with DAJAT [1], we employ the adversarial amplitude generator to produce 3 distinct adversarial amplitude spectra. These spectra are subsequently mixed with the amplitude spectrum of the benign sample. By combining the phase spectrum of the benign sample with each of these mixed amplitude spectra, we generate 3 recombined data samples.** These are considered as 3 augmentations of the original benign sample. For a more detailed explanation of the augmentation process, including formulas, please refer to the subsequent comment titled **(Continued) Re: Type of augmentations**.
>
> ---
>
> Using these recombined data, we conduct the experiments detailed in **Table A.1**, providing a comparison with DAJAT using 2 and 3 augmentations. Our results indicate that our DAT method offers a superior accuracy-robustness trade-off under identical experimental conditions.
>
> We hope this explanation clarifies your concerns. We greatly appreciate your valuable feedback and look forward to any further questions or comments you may have.
>
> ---

---

> > ### Comment · Reviewer_8ttu · 2024-08-13
> >
> > Thank the authors for the clarifications which have addressed my concerns. I have adjusted my rating accordingly.

---

> ### Author Response · Authors · 2024-08-13
> **(Continued) Re: Type of augmentations**
>
> To provide a clearer understanding of the statements "DAJAT [1] applies 2 and 3 data augmentations" and "we extend our DAT to include 2 and 3 augmentations", we offer a detailed explanation using a benign sample $(\mathbf{x},y)\in\mathcal{D}$ as an example.
>
> ---
>
> ### **For adversarial training with model $f$ using DAJAT [1], augmented data is generated as follows:**
>
> + **2 augmentations (#Aug. 2)**:
>
>     + Generate augmented data via AutoAugment:
>
>       $\mathbf{x}_1$=$\mathrm{AutoAugment}(\mathbf{x}),  \quad \mathbf{x}_2$=$\mathrm{AutoAugment}(\mathbf{x})$.
>
>     + Generate adversarial examples for $\mathbf{x}$,  $\mathbf{x}_1$, and $\mathbf{x}_2$ as $\mathbf{x}^{\prime}$,  $\mathbf{x}_1^{\prime}$, and $\mathbf{x}_2^{\prime}$, respectively.
>
>     DAJAT then uses $\mathbf{x}$,  $\mathbf{x}_1$, $\mathbf{x}_2$, $\mathbf{x}^{\prime}$,  $\mathbf{x}_1^{\prime}$, and $\mathbf{x}_2^{\prime}$ for training.
>
> + **Similarly,  DAJAT with 3 augmentations (#Aug. 3) follows the same procedure.**
>
> ---
>
> ### **For our DAT with IDFT $\mathcal{F}^{-1}(\cdot)$, model $f$, and adversarial amplitude generator $G$, recombined data  (which can be regarded as augmented data) is generated as follows:**
>
> + **2 recombined data (#Aug. 2)**:
>
>     + Generate adversarial amplitude:
>
>       $\mathcal{A_1}(\mathbf{x})=G (f(\mathbf{x}), \mathbf{z}_1),  \quad \mathrm{where}\ \mathbf{z}_1 \sim \mathcal{N}(0,1)$,
>
>       $\mathcal{A_2}(\mathbf{x})=G (f(\mathbf{x}), \mathbf{z}_2),  \quad \mathrm{where}\ \mathbf{z}_2 \sim \mathcal{N}(0,1)$.
>
>     + Obtain recombined data:
>
>       $\hat{\mathbf{x}}_1=\mathcal{F}^{-1}(\lambda_1 \cdot \mathcal{A_1}(\mathbf{x})+(1-\lambda_1)\cdot \mathcal{A}(\mathbf{x}),\mathcal{P}(\mathbf{x})), \quad \mathrm{where}\ \lambda_1\sim \mathrm{Uniform}(0,1)$,
>
>       $\hat{\mathbf{x}}_2=\mathcal{F}^{-1}(\lambda_2 \cdot {\mathcal{A_2}}(\mathbf{x})+(1-\lambda_2)\cdot \mathcal{A}(\mathbf{x}),\mathcal{P}(\mathbf{x})), \quad \mathrm{where}\ \lambda_2\sim \mathrm{Uniform}(0,1)$.
>
>     + Generate adversarial examples **with the same settings as DAJAT** for $\mathbf{x}$, $\hat{\mathbf{x}}_1$, and $\hat{\mathbf{x}}_2$ as $\mathbf{x}^{\prime}$, $\hat{\mathbf{x}}_1^{\prime}$, and $\hat{\mathbf{x}}_2^{\prime}$, respectively.
>
>     Our DAT then uses $\mathbf{x}$,  $\mathbf{x}_1$, $\mathbf{x}_2$, $\mathbf{x}^{\prime}$,  $\mathbf{x}_1^{\prime}$, and $\mathbf{x}_2^{\prime}$ for training, **following the settings of DAJAT**.
>
> + **Similarly,  DAT with 3 recombined data (#Aug. 3) follows the same strategy.**
>
> ---
>
> We hope this explanation provides greater clarity on the augmentations used in our approach. If there are any further questions or points that require clarification, we would be most grateful for your feedback. If our rebuttal has satisfactorily addressed your concerns, we humbly request that you consider reflecting this positively in your rating score.
>
> Thank you once again for your time, thoughtful consideration, and continued support.
>
> ---

---

> ### Author Response · Authors · 2024-08-13
>
> Dear Reviewer,
>
> We are truly grateful for your prompt response and for adjusting your rating. Your thoughtful feedback has been invaluable in refining our work, and we sincerely appreciate the time and effort you have dedicated to our submission. We will incorporate your suggested experiments and further clarifications in the final version.
>
> Thank you once again for your support.

---

### Author Rebuttal · Authors · 2024-08-06

Dear Reviewers and AC,

We sincerely appreciate the time and effort you have dedicated to evaluating our manuscript. The concerns and feedback raised during the initial review have significantly contributed to enhancing the quality of our paper. Below, we provide a summary of our key responses to the reviewers' suggestions and questions.

---

### **To Reviewer 8ttu**

1. **Performance Comparisons:**
   To address concerns regarding experimental results, we conduct additional experiments comparing the performance of our DAT framework with DAJAT [1] and [47].
   > See **R1** about DAJAT and See **R2** about [47] for Reviewer 8ttu.

2. **Complexity Analysis:**
   To clarify the complexity comparison between our DAT, DAJAT [1], and [47], we offer a detailed explanation of the hyperparameter settings and present the training time consumption analysis.
   > See **R3** to Reviewer 8ttu.

---

 ### **To Reviewer KucH**

1. **Feature Space Attacks:**
   In response to Reviewer KucH’s suggestions, we conduct experiments involving feature space attacks using various surrogate and target models. Detailed experimental comparisons are presented in **Tables 1-4** of the attached **rebuttal PDF**.
   > See **R1** to Reviewer KucH.

2. **Backbone Diversity:**
   As per Reviewer KucH’s recommendations, we perform experiments using a variety of backbones, including ViT, Inception, and DenseNet, and provide a comparative analysis between our DAT and existing methods.
   > See **R2** to Reviewer KucH.

3. **ImageNet-1K Experiments:**
   For ImageNet-1K, we conduct additional experiments to compare the performance of our DAT with current methods.
   > See **R3** to Reviewer KucH.

---

### **To Reviewer FTsi**

1. **Clarification on Figure 1:**
   Following Reviewer FTsi's suggestions, we further explain the impact of adversarial perturbations on phase and amplitude patterns. Additionally, we include more experimental results in **Figures 1-4** of the attached **rebuttal PDF**, demonstrating the influence of adversarial perturbations on both phase and amplitude, including their patterns and spectra.
   > See **A1** to Reviewer FTsi.

2. **Theoretical Analysis Clarification:**
   To clarify any confusion regarding learned features, we elaborate on our theoretical analysis, providing additional details and relevant citations.
   > See **A2** to Reviewer FTsi.

3. **Time Consumption Explanation:**
   To better illustrate the time consumption of DAT, we conduct experiments to provide a detailed analysis of the time and memory costs in comparison with other methods.
   > See **A3** to Reviewer FTsi.

4. **Discussion of Limitations:**
   In line with the suggestion, we outline additional limitations of DAT, which will be incorporated into the final version of the paper.
   > See **R3** to Reviewer FTsi.

---

---

### Author Response · Authors · 2024-08-13
**Request for Timely Reviewer Feedback Before Deadline**

Dear Area Chairs and Reviewers,

We sincerely appreciate the time and effort you and the reviewers dedicate to evaluating our submission. We are encouraged that **all reviewers view our work positively**. Following ***Reviewer FTsi***'s feedback, we clarify key technical aspects and conduct additional experiments, which are well received and **result in a score increase**. However, although ***Reviewer 8ttu*** has responded to our rebuttal, we are still awaiting a response to our latest comment, and ***Reviewer KucH*** has not yet responded to our rebuttal.

In response to ***Reviewer 8ttu***'s question regarding the **type of augmentations**, we offer further explanations with a detailed description of the augmentation approach used in DAJAT [1] and our DAT in **Re: Type of augmentations**, and a specific example in **(Continued) Re: Type of augmentations**.

For ***Reviewer KucH***, we expand our experiments to include **feature space attacks**, add **additional backbones like Inception_v3 and ViT-S**, and provide results on the **ImageNet-1K dataset**. These comprehensive experiments demonstrate the generalization and robustness of our method across different models and datasets, providing a thorough response to the reviewer's concerns about the scope and applicability of our approach.

As the rebuttal discussion period ends in **less than 24 hours**, ***Aug 13, 11:59 PM AoE***, we are concerned that we may not have sufficient time to address any potential additional feedback that may arise. We would be deeply grateful if you could kindly remind ***Reviewers 8ttu*** and ***KucH*** to review our responses at their earliest convenience. Your assistance in this matter would be greatly appreciated.

Thank you once again for your continued support and consideration.

Best regards,
Submission1111 Authors

---

### Decision · Program_Chairs · 2024-09-25

**Decision:**

Accept (poster)

**Comment:**

**Summary of the paper**

This paper proposes a new adversarial training method that improves adversarial robustness. Specifically, the authors focus on the frequency domain of adversarial perturbations and empirically verify that forcing the model to focus on the phase pattern would make the model more robust against adversarial attacks. Based on this observation, the authors propose an adversarial training strategy in which the model can learn the phase pattern unaffected by the adversarial attack. In particular, the authors suggest using a separate adversarial amplitude generator to keep the phase pattern unaffected by the adversarial attack. The authors provide some theoretical findings and experimental results regarding the proposed strategy.

**Summary of the discussion**

* In general, the reviewers mentioned similar strengths of the paper.
  * The paper is well-organized and easy to read.
  * Experiments show the improvement well.
* Reviewer 8ttu pointed out two problems with the authors’ method.
  * First, the experiment baselines are weak, and one existing method, DAJAT, performs better. Regarding the first point, the authors emphasized that the proposed method requires fewer augmentations per sample compared to other methods that the reviewer mentioned.
  * Also, the proposed method is so complicated that it might require extensive searching for hyperparameters. Regarding this concern, the authors clarified that their method uses only two hyperparameters, which is much simpler than the existing methods.
  * After the discussion, the reviewer changed the rating from 5 to 6.
* Reviewer KucH pointed out three problems with the authors’ method.
  * First, only decision space attacks are used for performance evaluation.
  * Second, the authors should have used more models for experiments.
  * Third, the experiments were not performed on the ImageNet dataset.
  * The authors referred to their existing results (in the Appendix) regarding all the points.
  * After the discussion, the reviewer stayed at the original rating of 5.
* Reviewer FTsi pointed out two problems.
  * First, the proposed method is too complicated. Regarding the first concern, the authors emphasize that the method contains only two modules to argue that the proposed method is simple enough.
  * Second, some descriptions could be clearer, and some details should be included. The reviewer asked for clarification of the missing details as Questions. Then, the authors answered all four questions from the reviewers, providing more references.
  * After the discussion, the reviewer changed the rating from 5 to 6

**Justification of my evaluation**

* My rating is 6.
* I agree with the strengths the reviewers mentioned. The experiments show improvement, and the performance looks convincing.
* The authors communicated with the reviewers effectively, and as a result, two reviewers increased their ratings.
* All the weaknesses that Reviewer KucH mentioned were covered in the Appendix, so the reviewer should have changed the rating. The reviewer told me it does not fully address the questions, but I disagree with the reviewer's point that the experiments were insufficient.

I’m confident about my evaluation and recommend accepting this work (poster).